# PAY-PER-SEARCH MODELS ARE ABSTENTION MODELS

## ABSTRACT

LLMs cannot reliably recognize their parametric knowledge boundaries and often hallucinate answers to outside-of-boundary questions. In contrast, humans recognize their limitations and can either seek external help for such questions or abstain. In this paper, we introduce MASH (**M**odeling **A**bstention via **S**elective **H**elp-seeking), a training framework that readily extracts abstentions from LLMs. Our key idea is that any external help-seeking by an LLM, i.e. search tool use, can serve as a proxy for abstention if the external help (search) is appropriately penalized while simultaneously rewarding answer accuracy. MASH operationalizes this idea using reinforcement learning with a pay-per-search reward.

We run experiments on three knowledge-intensive QA datasets. Our results show that MASH substantially improves upon the selective help-seeking performance of prior efficient search approaches; on multi-hop datasets, MASH improves answer accuracy by 7.6%. Furthermore, MASH demonstrates strong off-the-shelf abstention – it can distinguish between unanswerable/answerable questions and selectively generate responses for answerable questions – showcasing behavior analogous to specialized abstention approaches. We emphasize that contrary to prior abstention methods, MASH does not require pre-determining knowledge boundaries to construct training data. Instead, MASH's abstentions are a by-product of training for the auxiliary selective help-seeking task. Overall, we show that MASH training effectively aligns search tool use with parametric knowledge, which can be successfully leveraged for making abstention decisions.[1]

## 1 INTRODUCTION

A reliable AI assistant should recognize its knowledge boundaries – what questions it can and cannot effectively respond to – and act accordingly when a question is outside its boundaries. Conventionally, LLMs learn their knowledge boundaries through alignment by explicitly training for abstention (Yang et al., 2024; Cheng et al., 2024) and calibrated verbalization of uncertainty (Xu et al., 2024b; Stengel-Eskin et al., 2024). These strategies yield improved recognition of capability boundaries but are limited to reducing model errors. The number of questions a model can correctly answer remains unchanged. In this paper, we ask – can we design a training strategy that intrinsically yields an abstention model capable of recognizing its boundaries, while learning techniques that expand its set of answerable questions?

We look at human behavior for inspiration. Humans recognize their limitations and when asked for knowledge they cannot provide, either abstain or seek outside help. This external help-seeking can make otherwise unanswerable questions answerable. In this paper, we propose MASH (**M**odeling **A**bstention via **S**elective **H**elp-seeking), a framework that indirectly trains LLMs for abstention by instead training a model to engage in selective help-seeking, i.e. asking for help only when it cannot effectively respond to a query alone.

As a proof of concept, we explore this idea in the context of short-form question-answering tasks. We operationalize help-seeking as invoking a retrieval tool that returns information related to a given query. We train LLMs that selectively seek help (i.e. invoke retrieval) end-to-end with reinforcement learning using a pay-per-search penalty that discounts a correctness reward by the number of searches a model performs. An optimal policy optimizing this reward would, by definition, search only when a question cannot be reliably answered with parametric knowledge. In an inference mode with the same access to search, this model will mirror the above selective search behavior.

---

[1]We will publish code and model checkpoints upon acceptance.

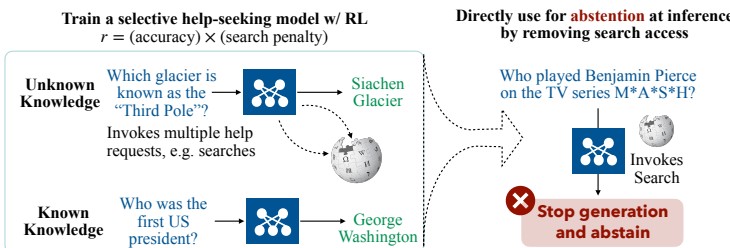

**MASH: M**odeling **A**bstention via **S**elective **H**elp-seeking

Figure 1: Overview of MASH's strategy for eliciting abstractions. Help-seeking LLMs are RL-trained to maximize answer accuracy while minimizing the searches. At inference, this same model is used for abstention by removing search access and treating any search requests as abstention.

But more importantly, we can readily elicit abstention decisions from this same model by removing its access to search tools – in that case, any search invocation serves as a proxy for abstention (see Figure 1). MASH, under this framing, effectively trains for two capabilities at the cost of one. Crucially, MASH assumes no privileged information regarding knowledge boundaries like standard abstention approaches (Yang et al., 2024; Cheng et al., 2024; Xu et al., 2024b) or require structured multi-agent interactions (Stengel-Eskin et al., 2024; Eisenstein et al., 2025).

We train MASH models using reinforcement learning with a pay-per-search reward (see Figure 1). However, baseline implementations of this idea (Wang et al., 2025a) result in efficient but sub-optimal search behaviors – models can converge to always searching at least once. To address this, we propose a lightweight synthetic data curation and SFT pipeline that, crucially, assumes no information about the LLM's parametric knowledge. Instead, it serves to inject diverse, albeit parametrically unaligned, search behavior in LLMs to improve exploration in later RL training. Additionally, we extend the reward formulations of prior work (Wang et al., 2025a) to obtain penalties with harsher levels of severity; this is crucial for extracting good help-seeking behaviors via RL.

We run our experiments on 3 different knowledge-intensive datasets, and evaluate both the selective help-seeking performance with regular inference (w/ access to search) and abstention performance (w/o access to search). Our results show that MASH models substantially outperform previous efficient search baselines (Wang et al., 2025a) at balancing answer accuracy and searches. Notably, on multi-hop datasets, MASH reports a 7.6% accuracy improvement with a better distribution of searches. In fact, this performance is on par with search baselines (Jin et al., 2025) that allow any number of searches (upto a max value) without any penalty. We investigate this further and show that this improvement can be attributed to MASH showcasing a broader range of search strategies, i.e. diversity over number of searches, as a direct result of its training recipe.

Furthermore, we show that MASH reports strong off-the-shelf abstention performance. It achieves competitive performance with our strongest abstention baseline DPO (Rafailov et al., 2023; Cheng et al., 2024), which explicitly constructs a specialized training dataset for abstention training. Moreover, compared to prompting and supervised training methods for abstention (Yang et al., 2024), MASH reports higher answer accuracy ($10 - 20\%$ improvement) over non-abstained questions by better differentiating between answerable/unanswerable questions.

Taken together, our results demonstrate that MASH is an effective technique that yields an abstention model capable of recognizing its boundaries, while simultaneously expanding its set of answerable questions via help-seeking.

## 2 MASH: Modeling Abstention via Selective Help-seeking

### 2.1 Abstention Framework

**Help-seeking LLMs** We assume an inference setting where a language model $\pi_\theta$ can ask for help by sending a help request $h$ to a helper $H(\cdot)$, which then returns a response $o \sim H(h)$. This helper $H$ can take various forms: it could be a tool such as a retrieval model responding to a query, another stronger language model or an actual human in-the-loop. The model would then condition on the response $o$ and continue its generation. Formally, given an input question $q$, the model samples a trajectory $\tau \sim \pi_\theta(\cdot|q; H)$ of the form $\tau = (r_1, h_1, o_1, \cdots, r_l, h_l, o_l, r_{l+1}, \hat{y})$, where

each $r_i$ represents reasoning, each $h_i$ represents a help request generated by $\pi_\theta$, $o_i$ represents the associated output from helper $H(\cdot)$ and $\hat{y}$ represents the model's final answer.

In this paper, we focus on knowledge-based domains. Here, $h_i$ is a search query generated by $\pi_\theta$, the helper $H(\cdot)$ is a retrieval model and $o_i$ is a set of top-$k$ documents retrieved by $H(h_i)$ from a document corpus. In practice, we assume that reasoning outputs $r_i$ are enclosed between <think> and </think>, search queries between <search> and </search>, and answers between <answer> and </answer> tokens. We use help/search, and helper/retriever interchangeably.

**Training Objective**  We want the language model $\pi_\theta$ to recognize its knowledge boundaries. We posit that we can obtain such a model – without privileged information regarding parametric knowledge boundaries – by training the model to maximize its accuracy while minimizing the number of search requests. Specifically, we optimize the following proxy objective:

$$\max_\theta \ \mathbb{E}_{(q,y)\sim D, \tau\sim\pi_\theta(\cdot|q;H)}[r_{acc}(y,\tau) \cdot r_{help}(q,\tau)] - \beta D_{KL}[\pi_\theta(\tau|q;H)||\pi_{\theta_{init}}(\tau|q;H)], \quad (1)$$

where $D$ is the dataset, $r_{acc}(y,\tau) \in \{0,1\}$ is a binary measure of correctness and $r_{help}(y,\tau) \in [0,1]$ is a multiplicative penalty that assigns a lower value the greater the number of searches in $\tau$. We use reinforcement learning, specially the GRPO algorithm (Guo et al., 2025), to optimize this objective.

**Eliciting Abstention from a Selectively Help-Seeking Model**  Let $\pi_{\theta*}$ be the optimal policy derived using the above objective. This model will selectively seek help – determine whether to answer a given question $q$ as a function of its expected parametric accuracy and the severity of the $r_{help}$ penalty. We re-frame the goal (and our subsequent evaluations) of this help-seeking model from efficiency, i.e. reducing number of searches, to parametric knowledge alignment, i.e. aligning search behavior with presence or absence of knowledge about a given question in the model's parameters.

Under this re-framing, we can readily elicit abstentions from a selectively help-seeking model by treating any search invocation as a proxy for abstention. Figure 1 illustrates this abstention framework, which we call MASH: **M**odeling **A**bstentions via **S**elective **H**elp-seeking.

## 2.2 Training a Selective Help-Seeking Model

MASH training involves two main steps: (1) initializing $\theta_{init}$ in Equation 1 such that it displays diverse search behaviors (zero, one, or multiple searches) to encourage exploration, and (2) a reward function that appropriately balances accuracy and search tool penalty.

### 2.2.1 Initializing $\pi_\theta$ w/ Warm-start SFT

RL training to optimize Equation 1 should, in theory, result in a model that selectively seeks help. However, in practice, we find that such training converges to sub-optimal policies – either exhibiting degenerate strategies that always or never search, or failing to learn to use the search tool effectively. In our work, we propose a **lightweight and model-agnostic synthetic data generation and finetuning pipeline** that results in a substantially better initial policy for subsequent RL training. Our data generation pipeline is designed to encourage diversity in the number of searches in model trajectories. Crucially, it requires no information about model's parametric knowledge boundaries. In fact, we bake this in explicitly by generating the synthetic fine-tuning dataset using a completely different model with different knowledge boundaries.

**Synthetic data generation**  Our overall algorithm is outlined in Algorithm 1. For each input question $q$ in the training dataset, we randomly sample a target number of searches $l$ for the associated trajectory and perform constrained decoding with the synthetic data generator $G$ to satisfy this constraint. We sample to generate $l$ consecutive thinking and search steps (appended with retrieved documents from retriever $H(\cdot)$). We achieve this by forcibly appending a <think> tag after the initial question and after retrieval outputs, and the <search> tag after the end of thinking tag </think>. We repeat this $l$ times. We sample multiple such trajectories per question, evaluate each and return a correct trajectory if one exists. Otherwise, we return the trajectory with the shortest answer. Note that this constrained decoding process is only used during synthetic data generation.

A warm start SFT step is also included in recent works' training pipelines to improve subsequent RL training (Guo et al., 2025; Gandhi et al., 2025; Wang et al., 2025b). However, we highlight one key difference. Contrary to prior works, our warm start process does not not target correctness or alignment with model's parametric knowledge – the two central goals of MASH. In fact, our synthetic data contains $35\%$ errors with respect to answer correctness and, by design, yields a policy

whose search behavior is unaligned with its parametric knowledge (discussed in Appendix C.3). The model learns how and when to use searches during RL training.

---

**Algorithm 1** Warm-Start Trajectory Construction

---

**Input:** Datapoint $(q, a^*)$, generator $G$, retriever $H$, maximum searches $l_{max}$, num samples $N$
**Output:** Datapoint $(q, \tau)$ for SFT
    Sample random number of searches $l \sim \{0, \ldots, l_{max}\}$
    Define seq $\leftarrow$ [**think**, **search**] $\times l +$ [**think**, **answer**]
    **for** $i = 1 \rightarrow N$ **do**
        Initialize current trajectory $\tau_i \leftarrow \emptyset$
        **for** action in seq **do**
            Append action start tag $\tau_i \leftarrow \tau_i + <$ action $>$
            Generate action $a \sim G(\cdot|q, \tau_i)$ until $</$action$>$
            Append action $a$ to trajectory $\tau_i \leftarrow \tau_i + a$
            **if** action $=$ search **then**
                Retrieve top-$k$ documents $o \sim H(a)$ and append to trajectory $\tau_i \leftarrow \tau_i + o$
    Set $\tau$ to a random correct $\tau_i$ if any, else $\tau_i$ with shortest answer.
    **return** $\tau$

---

### 2.2.2 REWARD FORMULATION

Our reward $r(y, \tau)$ is a product of two terms: $r_{acc}(y, \tau)$, which is a binary correctness reward and $r_{help}(y, \tau)$, which is a search tool penalty. We compute $r_{acc}(y, \tau)$ using exact match.

The form and severity of $r_{help}$ will influence the learned help-seeking behavior. For input question $q$ and $G$ output trajectories $\{\tau_i\}_{i=1}^{G}$ sampled during GRPO, let $n$ be the number of search queries in the most efficient and correct trajectory $\tau^{\text{ef}}$ and $m$ be the number of queries in the given trajectory $\tau_i$. We want $r_{help}$ to appropriately penalize $\tau_i$ if $m > n$. There exists an arbitrarily high number of penalty formulations that satisfy this desiderata; we experiment with three:

1. **Exponential Decay**, defined as $r_{help}^{\text{EXP}}(q, \tau_i) = \lambda^{m-n}$ where $\lambda$ controls the severity of the penalty.

2. **OTC** reward proposed by Wang et al. (2025a). We follow their recommendation and set $c$ to the maximum number of searches allowed in a single trajectory.

$$r_{help}^{\text{OTC}}(q, \tau_i) = \begin{cases} 1 & \text{if } m = n = 0 \\ \cos(\frac{m \cdot \pi}{2m+c}) & \text{if } n = 0 \\ \sin(\frac{m \cdot \pi}{m+n}) & \text{otherwise} \end{cases}, \quad (2)$$

3. **OTC-Strict** which enforces an extremely strict tool use penalty when $m > n = 0$. Note that $n = 0$ indicates there is a correct trajectory $\tau^{\text{ef}}$ without any searches. We posit that for these cases, any other trajectory $\tau_i$ that uses searches should get a 0 reward under a very strict definition of answerability. Therefore, we set $r_{help}^{\text{OTC-St}}(q, \tau_i)$ to 0 for such cases. We can use any of the above two reward formulations for when $n > 0$, but choose OTC's sinusoidal function to align with prior work.

## 3 EXPERIMENTAL SETUP

**Datasets and Models** We run our experiments on three knowledge-intensive datasets – the single-hop dataset Natural Questions (NaturalQA) (Kwiatkowski et al., 2019), and multi-hop datasets Hot-PotQA (Yang et al., 2018) and 2WikiMultiHopQA (2Wiki) (Ho et al., 2020).[2] We train and evaluate on each dataset separately; this allows us to evaluate MASH across tasks requiring different search strategies and with different distributions of parametrically answerable questions. We perform all training and evaluation on the Qwen2.5-3B base model (Qwen et al., 2025).We deliberately choose the base model over instruct as the latter has already undergone abstention training although the exact training strategy is unknown; we propose MASH as an alternative. We use the E5 retriever (Wang et al., 2022) and the 2018 Wikipedia dump as our knowledge source (Karpukhin et al., 2020).

**Hyperparameters** For the OTC reward, we follow Wang et al. (2025a) and set $c$ equal to the maximum number of searches. For Exponential Decay, we set $\lambda$ to $0.5$ for Natural Questions and $0.8$

---

[2]We find that the "comparison" and "bridge-comparison" questions comprising in 2WikiMultiHopQA have unbalanced answer distributions (skewed towards "no"). This opens up the possibility of reward hacking by exploiting this dataset property. Therefore, we omit these questions from our training and evaluation.

otherwise. We note that these hyperparameter choices imply the following decreasing order of severity of search penalty: OTC-STRICT→EXP→OTC. For each search query, we fix the response to be the top-3 retrieved passages and allow a maximum of 5 searches per trajectory. We use the veRL library (Sheng et al., 2025) for RL training. More training details are in Appendix C.1.

**Warm-start data generation** We follow the strategy outlined in Section 2.2.1 to generate warm-start data for each dataset using Qwen2.5-32B base. This ensures that information about knowledge boundaries is not baked into the SFT training data and that samples follow the prescribed format. For each dataset, we randomly sample 1000 questions from its training set and set $l_{max} = 2$. We select the trajectory for each question from $N = 5$ samples. Details can be found in Appendix C.3

We evaluate our selective help-seeking models in two inference modes: (1) **w/ access to search tools**, which directly aligns with its training, and (2) **w/o search tools**, where we use the help-seeking model for abstention. The baselines and evaluation metrics for these are described next.

## 3.1 EVALUATION DETAILS FOR INFERENCE MODE I: W/ SEARCH TOOLS

**Baselines** We compare MASH's help-seeking model against the following baselines that also conduct RL training, but with different setups: (1) `R1` trained using RL but without access to any search tools during training or evaluation. This baseline provides an upper bound for answer accuracy using only parametric knowledge. (2) `Search-R1` (Jin et al., 2025) trained w/ search tools and a binary correctness reward; showcasing an upper bound without any penalties for searching, (3) `OTC` (Wang et al., 2025a) RL-trained for efficient search tool use. We compare these baselines to three MASH variants that differ in reward penalties (refer to § 2.2.2). Note that MASH w/ OTC and OTC differ in the warm-start procedure applied to the former.

**Evaluation Metrics** We want our help-seeking model to strike a balance between answering parametrically (w/o search calls) and seeking help (w/ search calls). We report three metrics that collectively capture this: (1) **Accuracy (Acc)**, i.e. if the predicted answer matches the gold response. Due to the limitations of exact match, we use an LLM judge, namely DeepSeek-V3.1 (Liu et al., 2024), to determine this. (2) **Tool calls (TC)**, i.e. the average number of searches across trajectories. (3) **Tool Productivity (TP)** (Wang et al., 2025a), which is defined as $[\sum_{i=1}^{|\mathcal{D}|} \mathbb{I}\{y_i = \hat{y}_i\}/(1 + m_i)]/|\mathcal{D}|$ for test set $\mathcal{D}$. This discounts the accuracy of each output trajectory by its number of searches $m_i$. For all models, we report these metric averages over 4 samples. We use TP on the validation set to select our model checkpoints for all methods, except Search-R1 for which we use accuracy; TP will result in a much inferior checkpoint selection for this case.

## 3.2 EVALUATION DETAILS FOR INFERENCE MODE II: ABSTENTION

In this evaluation mode, we follow the MASH process outlined in Figure 1 and § 2.1 to extract abstentions from a help-seeking model by removing access to search tools at inference.

**Baselines** We compare against the following abstention baselines: (i) **5-shot prompting** with the base model, with abstention/not of in-context exemplars decided based on its parametric knowledge. (ii) **Alignment for Honesty - Absolute** (AFH-Abs) (Yang et al., 2024), which does SFT on a specially curated abstention dataset by pairing each input question with either the output "I abstain" or the gold answer, depending on the base model's knowledge boundaries. (iii) **Alignment for Honesty - Multisample** (AFH-Mult) (Yang et al., 2024), which constructs multiple training samples for each question, pairing it with either "I abstain" or the gold answer depending on the average correctness over multiple outputs, for SFT training. (iv) **DPO**, inspired by Cheng et al. (2024), which pairs each question with a preferred and dispreferred output. If the question is parametrically answerable, we set these to be the gold answer and "I abstain" respectively; this is switched for parametrically unanswerable questions. We train with the DPO loss objective (Rafailov et al., 2023) and SFT loss added as a regularizer (Pang et al., 2024).

Each of (1), (2) and (3) requires a definition of answerability; i.e. when can we claim that a question is answerable. A standard technique is to estimate the accuracy over 10 samples and use a threshold $\lambda$ to classify into answerable/not. However, there does not exist a consensus in prior works on how to decide this threshold (Yang et al., 2024; Chen et al., 2024). In our paper, we follow Yang et al. (2024) and set $\lambda = 0.1$. Exact data curation and training details are in Appendix D.

**Evaluation Metrics** For abstention evaluation, we report two kinds of metrics: (i) **Answer Accuracy**: We report overall accuracy, i.e. over the entire test set, and precision, i.e. over non-abstained

| Method | Natural Questions | | | HotPotQA | | | 2Wiki | | |
|---|---|---|---|---|---|---|---|---|---|
| | Acc↑ | TC↓ | TP↑ | Acc↑ | TC↓ | TP↑ | Acc↑ | TC↓ | TP↑ |
| R1 | 26.06 | 0.0 | 26.06 | 26.54 | 0.0 | 26.54 | 9.17 | 0.0 | 9.17 |
| Search-R1 (Jin et al., 2025) | 57.29 | 1.0 | 28.65 | 56.36 | 3.00 | 14.09 | 45.36 | 3.00 | 11.34 |
| OTC (Wang et al., 2025a) | 58.95 | 1.0 | 29.47 | 44.76 | 0.81 | 28.64 | 39.59 | 1.57 | 15.32 |
| MASH w/ OTC | 59.83 | 1.0 | 29.97 | 55.42 | 1.14 | **32.91** | 45.99 | 1.6 | 18.87 |
| MASH w/ OTC-ST | 56.40 | 0.64 | **38.64** | 53.34 | 1.10 | 32.55 | 46.23 | 1.64 | **19.08** |
| MASH w/ EXP-DEC | 54.31 | 0.65 | 36.59 | 53.79 | 1.07 | 32.10 | 44.29 | 1.53 | 18.09 |

Table 1: Accuracy, average number of tool calls (TC) and tool productivity (TP) statistics for baselines and MASH evaluated under **inference w/ search tools**. MASH w/ OTC-ST is our best model with a $4.22\%$ and $5.61\%$ mean improvement on Acc and TP resp. over baseline OTC across datasets.

questions. Note that over-conservativeness, i.e. aggressively abstaining, will hurt overall accuracy but increase precision, while under-conservativeness will have the opposite effect. (ii) **Abstention Classification**: This captures whether a model's abstention behavior is aligned with its knowledge boundaries, agnostic of answer accuracy. To avoid defining answerability (different reward penalties assume a different answerability threshold), we evaluate over two groups of questions unaffected by the choice of $\lambda$, i.e. questions that the base models always answer incorrectly or always correctly. Let $\%\text{Abs}(0)$ and $\%\text{Abs}(1)$ be the percentage of questions for which a model abstains for the above two groups, respectively. We report $\%\text{Abs}(0)$ and Delta ($\%\text{Abs}(0) - \%\text{Abs}(1)$). A model that recognizes its knowledge boundaries should have a high abstention rate for always incorrect questions, i.e. $\%\text{Abs}(0)$, and a much lower abstention rate for always correct questions, captured by a large margin $\%\text{Abs}(0) - \%\text{Abs}(1)$. We do not evaluate the 2Wiki dataset for abstention classification due to there being only 58 test examples in the $\text{Abs}(1)$ bucket, preventing reliable conclusions.

## 4 RESULTS

### 4.1 INFERENCE MODE I: W/ SEARCH TOOLS

We first evaluate the performance of baselines and MASH in the inference setting with access to search tools. Table 1 reports overall answer accuracy, average tool calls and tool productivity for all methods. Additionally, we show the distribution of tool calls (TC=0/1/2+) and the corresponding accuracy per search count (subscript) in Table 2. This allows us to conduct an apples-to-apples comparison between models' accuracy for the same number of tool calls.

**MASH outperforms all search baselines on tool productivity by effectively balancing accuracy and searches.** Our results in Table 1 show that MASH, particularly MASH w/ OTC-Strict, leads to a 5.61 point improvement on tool productivity over baseline OTC on average across datasets. Surprisingly, MASH variants report accuracies on par with Search-R1 (trained without any tool use penalty) on multi-hop datasets HotPotQA and 2Wiki, but with a substantially lower number of searches (1.64 vs 3). Moreover, this performance is a massive improvement over baseline OTC (∼10% and ∼4% improvements on HotPotQA and 2Wiki respectively) with only a slightly higher number of searches. Tool productivity, which accounts for both these metrics, improves by 4 points on average over baseline OTC. Taken together, these results suggest that MASH not only reduces the average number of searches, but also better operationalizes them to maintain accuracy.

**Severe search penalties are needed for parametric answers for single-hop NaturalQA.** We observed that both baseline OTC and MASH with the lenient OTC penalty (MASH w/ OTC) do not learn to answer parametrically for NaturalQA, i.e. converge to TC=1 for all questions. On the other hand, MASH w/ OTC-Strict answers parametrically for $36\%$ of the questions with only a $2.5\%$ drop in accuracy, thereby improving tool productivity by 9 points. Similarly, MASH w/ Exp-Dec answers parametrically $35\%$, with a $4.5\%$ drop in accuracy[3] compared to baseline OTC but a 7 point improvement in tool productivity.[4]

---

[3]Note that MASH w/ Exp-Dec training did result in checkpoints with higher accuracies. However, we use tool productivity on the validation set as the metric to select the final checkpoint for all methods.

[4]The multi-hop datasets, HotPotQA and 2Wiki, report slightly higher average tool calls with the strictest penalty (MASH w/ OTC-Strict), presumably contradicting the above claim. However, fine-grained search distributions (see Table 2) show that, similarly to NaturalQA, OTC-Strict does answer parametrically (TC=0) more often than the lenient versions. The increase in average tools calls is due to a larger fraction of 2 searches.

| Method | Natural Questions | | | HotPotQA | | | 2Wiki | | |
|---|---|---|---|---|---|---|---|---|---|
| | 0 | 1 | 2+ | 0 | 1 | 2+ | 0 | 1 | 2+ |
| OTC | $0.0_{0.0}$ | $100.0_{58.9}$ | $0.0_{0.0}$ | $19.5_{64.5}$ | $80.2_{40.0}$ | $0.3_{32.0}$ | $3.1_{24.1}$ | $36.7_{26.6}$ | $60.2_{48.3}$ |
| MASH w/ OTC | $0.2_{53.6}$ | $99.8_{59.8}$ | $0.0_{33.3}$ | $23.5_{66.5}$ | $41.7_{58.2}$ | $34.8_{44.6}$ | $13.0_{31.3}$ | $13.9_{35.9}$ | $73.1_{50.5}$ |
| MASH w/ OTC-ST | $36.4_{57.4}$ | $63.5_{55.9}$ | $0.1_{17.6}$ | $28.9_{59.9}$ | $34.7_{56.4}$ | $36.4_{45.2}$ | $14.3_{32.5}$ | $8.3_{42.3}$ | $77.5_{49.2}$ |
| MASH w/ EXP-DEC | $35.2_{53.6}$ | $64.8_{54.7}$ | $0.0_{20.0}$ | $23.7_{64.0}$ | $45.5_{53.4}$ | $30.8_{46.5}$ | $11.8_{32.1}$ | $23.4_{20.6}$ | $64.8_{55.0}$ |

Table 2: Fine-grained tool use distribution (TC=0/1/2+ search) for baseline OTC and MASH models. We also report answer accuracies for questions in each subset (subscript). TC=0 indicates that the model answers parametrically. MASH can successfully off-load questions to parametric answering (from TC=1 to TC=0) will minimal or no decrease in accuracy (HotPotQA & NaturalQA).

| Method | Answer Accuracy | | | | | | Abstention Classification | | | |
|---|---|---|---|---|---|---|---|---|---|---|
| | NaturalQA | | HotPotQA | | 2Wiki | | NaturalQA | | HotPotQA | |
| | Acc | Prec | Acc | Prec | Acc | Prec | Abs(0) ↑ | Delta↑ | Abs(0) ↑ | Delta↑ |
| OTC | 0.0 | 0.0 | 12.6 | 64.5 | 0.75 | 24.1 | 100.0 | 0.0 | 95.3 | 41.4 |
| MASH w/ OTC | 0.1 | 31.1 | 15.6 | 66.5 | 4.1 | 31.3 | 99.9 | 0.1 | 94.8 | 52.3 |
| MASH w/ OTC-ST | 20.9 | 57.4 | 17.3 | 59.9 | 4.6 | 32.5 | 85.5 | 66.2 | 91.2 | 60.3 |
| MASH w/ EXP | 18.9 | 53.6 | 15.2 | 64.0 | 3.8 | 32.2 | 85.7 | 62.7 | 94.5 | 52.7 |
| 5-shot Prompting | 23.4 | 42.5 | 14.7 | 31.5 | 3.6 | 10.9 | 60.2 | 44.6 | 60.5 | 26.9 |
| AFH (Absolute) | 21.7 | 43.3 | 20.7 | 34.2 | 4.7 | 18.5 | 67.7 | 48.1 | 50.4 | 35.4 |
| AFH (Multisample) | 14.7 | 54.8 | 12.9 | 53.8 | 2.6 | 29.2 | 87.9 | 52.1 | 89.2 | 57.6 |
| DPO | 22.3 | 56.2 | 19.9 | 53.1 | 3.3 | 31.6 | 84.5 | 71.6 | 85.9 | 73.5 |

Table 3: Abstention accuracy (left) and abstention classification (rights) results for specialized abstention approaches and MASH. For abstention accuracy, we report overall Acc over the entire test set and Prec, i.e. accuracy over the non-abstained answers for each method. For classification, we report Abs(0), i.e. % abstention for unanswerable questions (higher better), and the delta (higher better) between the % abstention between unanswerable and answerable questions.

**MASH variants extract better and more diverse search behaviors for multi-hop datasets via RL.** Comparing search statistics for MASH w/ OTC and baseline OTC in Table 2, we see that they report a comparable number of parametric answers (23.5% vs 19.5%) but show very different search behaviors for the remaining questions. Particularly, the baseline OTC model without warm-start collapses to only one search for the remaining 80.2% of its trajectories, while the warm-started model (MASH w/ OTC) can perform a mixture of one and multi-hop searches. In fact, MASH variants report a much higher accuracy for one search questions (56.4% vs 40.0%) by offloading the more "difficult" questions, i.e. those the model cannot answer with only one search, to the two search bucket. Baseline OTC fails to do this and reports lower overall accuracy. We see similar trends for the other multi-hop dataset, 2Wiki, as well.

**MASH successfully aligns search tool use with parametric knowledge.** For NaturalQA, the fine-grained search statistics in Table 2 show that the the questions that MASH w/ OTC-Strict and w/ Exp-Dec answer parametrically have similar answer accuracy compared to those for which they invoke one search call (57.4 vs 55.9 for w/ OTC-Strict). This clearly shows that MASH can distinguish between parametrically answerable and not answerable questions and preferentially invoke tool calling for the latter to maintain overall accuracy.

### 4.2 INFERENCE MODE II: W/ ABSTENTION

**MASH shows strong abstention behavior off-the-shelf.** Tables 3 (left) reports the answer accuracy for the overall test dataset (Acc) and the non-abstained questions (Prec) for each method.[5] First, we observe that, apart from MASH w/ OTC on NaturalQA, all MASH variants substantially outperform the prompting and Alignment for Honestly based SFT approaches in terms of answer precision and report comparable overall accuracy. In a couple of instances, we find that the AFH (Absolute) baseline reports better accuracy (e.g. HotPotQA and NaturalQA) compared to MASH, but this accompanied by a 10-20% drop in precision.

---

[5]Note that it is possible to game one of these metrics by being over- or under-conservative. Therefore, all our conclusions are based on analyzing the two metrics collectively.

We find that MASH w/ OTC-Strict, our best performing model from Section 4.1, is comparable to DPO for NaturalQA and HotPotQA; it outperforms DPO based on Prec. (59.89 vs 53.14 for HotPotQA) but reports lower overall accuracy (17.33 vs 19.9). We attribute this to the fact that MASH w/ OTC-Strict is more conservative, i.e. more likely to abstain, than DPO. For 2Wiki, MASH w/ OTC-Strict outperforms DPO on both Acc and Prec metrics.

**MASH can differentiate between answerable and unanswerable questions.** Table 3 (right) shows the abstention classification results. As expected, we find that DPO models explicitly trained for abstention report the best results. Encouragingly, we see that MASH variants, except MASH w/ OTC on NaturalQA which does not learn to answer parametrically, report similarly high Abs(0) percentages as DPO. While DPO reports higher Delta for both datasets, Table 3 shows that these large improvements in Delta are often accompanied by a drop in precision. For e.g, DPO reports 13.17% better Delta than MASH w/ OTC-Strict for HotPotQA, but reports a 6.75% lower precision.

Taken together, these results present an encouraging picture for the idea of modeling abstention with models trained for the auxiliary selective help-seeking task. They show that although MASH does not train explicitly for abstention, its **abstention behavior is analogous to that of abstention methods leveraging oracle information regarding model knowledge boundaries.**

### 4.3 ANALYSIS 1: IMPACT OF WARM-START ON MASH PERFORMANCE

The comparative results of OTC baseline and MASH w/ OTC in both Tables 1 and 2 indicate that the warm-start SFT training is key to MASH's success. By design, it enables the model

| Method | Natural Questions | | | HotPotQA | | | 2Wiki | | |
|--------|------|------|------|------|------|------|------|------|------|
| | Acc↑ | TC↓ | TP↑ | Acc↑ | TC↓ | TP↑ | Acc↑ | TC↓ | TP↑ |
| OTC | 58.95 | 1.0 | 29.47 | 44.76 | 0.81 | 28.64 | 39.59 | 1.57 | 15.32 |
| OTC-ST | 52.34 | 0.49 | 39.28 | 26.99 | 0.0 | 26.99 | 10.41 | 0.0 | 10.41 |
| EXP | 57.58 | 1.00 | 28.79 | 41.48 | 0.71 | 28.68 | 9.71 | 0.0 | 9.71 |

Table 4: MASH **w/o warm-start** tested in **inference w/ search** mode.

to explore diverse trajectories with varying numbers of search tool calls during RL. Here, we study the impact of warm start for all reward formulations. Table 4 reports the performance for all three w/o warm start (refer to Table 1 for comparison with models trained w/ warm start).

**Warm-start adds stability to harsher penalties.** The OTC reward shows the best help-seeking behavior when considering all datasets collectively. However, we discussed in § 4.1 that the search behavior w/ warm-start is far superior to w/o for OTC. Recall that Exponential Decay and OTC-Strict both impose harsher penalties on search tool use than OTC. We observe that this results in severe training instabilities for these two when trained without warm-start – HotPotQA policy collapses to zero searches for OTC-Strict and the 2Wiki policy collapses for both Exponential Decay and OTC-Strict. Warm-start SFT, however, enables both to have successful training runs on all datasets, with OTC-Strict w/ warm start even substantially outperforming OTC in all evaluation modes.

### 4.4 ANALYSIS II: DO ORACLE HELPERS IMPROVE SELECTIVE HELP-SEEKING LLMS?

All experiments in Section 4 rely on a retrieval model (E5; Wang et al. (2022)) as the helper $H(\cdot)$. However, search results output by these retrievers can be noisy, which in turn generates a noisy signal for training the selective help-seeking LLM via RL. This prompts us to investigate if improving the "helper", as opposed to the reward or initialization, can improve the learned help-seeking behavior. **Setup:** We set $H(\cdot)$ to be an oracle; it directly returns the gold answer if the LLM invokes a help tag in its trajectory (exact prompts used is included in Appendix E). We train all MASH variants (OTC, OTC-Strict, Exp) for all datasets. Warm-start training is done for each individually with $l_{max} = 1$.

**Results: Help-seeking with oracle helpers fails to yield abstention behaviors.** We find that every single training run converged to always asking for help within the first 50 training steps , even for the stricter help penalties. Note that the optimal policy should display selective help-seeking, i.e. answer parametrically for known questions, in order to maximize the chosen reward. However, we do not observe this in practice, as always seeking-help is an easy strategy for the LLMs to discover. For OTC and Exponential Decay, it is given non-zero rewards for all inputs. For OTC-Strict, it is given a positive reward for each question without correct parametric answers, which will be common early in training. This shows that the noisiness of the retrieval model is crucial to extract selective help-seeking over training, in a manner aligned with its parametric knowledge.

| Method | Natural Questions | | | | TriviaQA | | | |
|---|---|---|---|---|---|---|---|---|
| | Acc↑ | Acc w/ tool↑ | Abs(0) ↑ | Delta↑ | Acc↑ | Acc w/ tool↑ | Abs(0) ↑ | Delta↑ |
| OTC | 2.1 | 54.36 | 99.04 | 8.32 | 4.07 | 71.43 | 96.95 | 7.11 |
| MASH w/ OTC-ST | 18.25 | 51.24 | 79.94 | 51.62 | 30.52 | 67.61 | 77.53 | 51.55 |
| DPO | 24.4 | - | 77.38 | 68.23 | 41.6 | - | 71.71 | 66.23 |

Table 5: Out-of-distribution accuracy (w/ and w/o search) and abstention classification results for baseline OTC, best MASH, and best abstention models trained on HotPotQA .

Note that **this setting with the oracle helper is equivalent to explicitly training for abstention using RL**, with decreasing magnitude of rewards assigned for correct answers, abstention and incorrect answers. All training runs collapsing to always seeking help indicates that abstention training setting would also fail. We require RL algorithms with better exploration to succeed in this setting.

### 4.5 ANALYSIS III: OUT-OF-DISTRIBUTION PERFORMANCE

Finally, we evaluate our trained models out-of-distribution. Due to space, we restrict our analysis to the OTC baseline, and our best performing MASH variant w/ OTC-Strict and the best abstention baseline (DPO) trained on HotPotQA. We evaluate generalization to other training datasets and an additional single-hop dataset TriviaQA (Joshi et al., 2017).

**Results:** Table 5 reports our results (NaturalQA and 2Wiki models are in Appendix F). MASH generalizes better than the OTC (higher Accuracy and Delta values), which abstains on nearly all questions out-of-distribution. MASH also reports better Abs(0) performance that DPO but lower Delta. We attribute this to MASH generalizing more conservatively out-of-domain. With 2Wiki, which exclusively contains two-hop questions, MASH generalizes relatively well to HotPotQA but fails on single-hop datasets. We argue that, under poor out-of-distribution accuracy generalization, abstention and invoking search tools is the more ideal decision. With search enabled, our HotPotQA-trained MASH model attains $24.43\%$ higher accuracy than DPO, which is limited to abstention.

## 5 RELATED WORK

**Abstention and Verbalized Uncertainty** Past work has explored developing techniques for hallucination detection (Du et al., 2024; Chen et al., 2024), abstention (Yang et al., 2024; Cheng et al., 2024) and calibration (Kapoor et al., 2024), with methods ranging from prompting (Feng et al., 2024) and hidden state probing (Du et al., 2024; Chen et al., 2024) to training of the model itself (Kadavath et al., 2022). For abstention, past work primarily uses pipelined approaches that first estimate a model's knowledge boundaries and then use this information either to construct datasets for SFT (Yang et al., 2024; Zhang et al., 2024) and DPO training (Cheng et al., 2024), train model-specific reward functions for RLHF (Xu et al., 2024a), or summarize uncertainty over multiple samples (Xu et al., 2024b). Alternative strategies featuring structured, multi-agent interaction scenarios (Stengel-Eskin et al., 2024; Eisenstein et al., 2025) have also been recently proposed. **Selective RAG** Separately, there has been explorations into developing retrieval augmented generation (RAG) approaches that know when to search or continue searching; these rely on uncertainty estimation through operations on hidden model states (Yao et al., 2025; Baek et al., 2025), self-consistency over samples (Ding et al., 2024) or output probabilities (Jiang et al., 2023; Su et al., 2024). We focus on knowledge-intensive queries but our approach is task-agnostic and only involves end-to-end RL. **Augmenting LLMs with Tool-Use** Recent works have proposed leveraging tool-use to augment LLM capabilities (Schick et al., 2023; Yao et al., 2023), with post-training pipelines for foundation models (Yang et al., 2025; Team et al., 2025) increasingly featuring dedicated training for tool-use. We build on top of recent work that trains LLMs to use search tools with RL (Jin et al., 2025), particularly on top of the OTC reward formulation of Wang et al. (2025a).

## 6 CONCLUSION

We propose MASH, a novel framework that trains LLMs for selective help-seeking, and readily extracting abstention behaviors. MASH trains models for two capabilities at the cost of one – models learn how to use search tools and synthesize information, and distinguish between answerable/unanswerable questions. Our results on 3 short-form knowledge-intensive datasets show that MASH outperforms previous efficient search baselines on overall accuracy when allowed searches and also demonstrates strong abstention behaviors, analogous to specialized abstention methods.

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

# A    THEORETICAL ANALYSES

## A.1    PRELIMINARIES

In this section, we will provide a theoretical analysis of the search behavior the optimal policy for the proxy objective of Equation 1 will display for a given question. Specifically, we will formally demonstrate that the optimal policy will produce a parametric answer to a question $q$ if and only if its expected reward when answering $q$ parametrically is greater than or equal to its expected reward when performing searches.

Before starting the analysis proper, we will make two trivial assumptions.

- **Assumption 1:** We set the KL penalty weight $\beta = 0.0$ to simplify the equation and focus solely on the search behavior of an optimal policy maximizing reward.

- **Assumption 2:** We assume that the optimal policy cannot achieve perfect expected accuracy, i.e. across multiple samples, for all questions when answering parametrically. This is because, under our training setup, a policy cannot acquire new knowledge beyond the base model's knowledge boundaries during RL. The purpose of training is instead to align its search behavior with its knowledge.

## A.2    ANALYSIS

Let $N_s(\tau)$ be the number of search calls made by a trajectory $\tau$ and let $\theta^*$ be the parameters optimizing the proxy objective. We will then prove the following claim:

**Claim:**    For a given question-answer pair $(q, y)$, the optimal policy may answer the question $q$ parametrically, that is without any searches, if and only if

$$E_{\tau \sim \pi_{\theta^*}(\cdot|q,H)}[r_{acc}(y,\tau) \cdot r_{help}(q,\tau)|N_s(\tau) = 0] \geq E_{\tau \sim \pi_{\theta^*}(\cdot|q,H)}[r_{acc}(y,\tau) \cdot r_{help}(q,\tau)|N_s(\tau) = i]$$

for all $i > 0$.

**Proof:**    Let $\theta^*$ be the set of parameters that optimizes the proxy objective and consider an arbitrary question-answer pair $(q, y)$. We firstly claim that the optimal policy's expected reward given this question $q$ can be written as a weighted sum of its expected reward when answering question $q$ with different search counts. We show this below:

$$E_{\tau \sim \pi_{\theta^*}(\cdot|q,H)}[r_{acc}(y,\tau) \cdot r_{help}(q,\tau)] = \Sigma_\tau \left[ \pi_{\theta^*}(\tau|q,H) \cdot r_{acc}(y,\tau) \cdot r_{help}(q,\tau) \right]$$

$$= \Sigma_{i=0}^\infty \Sigma_{\tau|N_s(\tau)=i} \left[ \pi_{\theta^*}(\tau|q,H) \cdot r_{acc}(y,\tau) \cdot r_{help}(q,\tau) \right]$$

$$= \Sigma_{i=0}^\infty \left[ P(N_s(\tau) = i|q) \cdot \Sigma_{\tau|N_s(\tau)=i} \left[ \frac{\pi_{\theta^*}(\tau|q,H)}{P(N_s(\tau) = i|q)} \cdot r_{acc}(y,\tau) \cdot r_{help}(q,\tau) \right] \right]$$

$$= \Sigma_{i=0}^\infty \left[ P(N_s(\tau) = i|q) \cdot E_{\tau \sim \pi_{\theta^*}(\cdot|q,H)} \left[ r_{acc}(y,\tau) \cdot r_{help}(q,\tau)|N_s(\tau) = i \right] \right],$$

where $P(N_s(\tau) = i|q)$ is the probability that the optimal policy will produce a trajectory $\tau$ with $N_s(\tau) = i$ given question $q$.

Given this, we will first prove the forward direction. Assume that the optimal policy answers question $q$ parametrically. This means that $P(N_s(\tau) = 0|q) = 1$. Assume for the sake of contradiction that there exists some $i$ such that

$$E_{\tau \sim \pi_{\theta^*}(\cdot|q,H)}[r_{acc}(y,\tau) \cdot r_{help}(q,\tau)|N_s(\tau) = 0] < E_{\tau \sim \pi_{\theta^*}(\cdot|q,H)}[r_{acc}(y,\tau) \cdot r_{help}(q,\tau)|N_s(\tau) = i]$$

Then, we can construct a different policy $\hat{\theta}$ that has the same distribution for all other questions $\hat{q}$ but always answers with $i$ searches for question $q$. This policy $\hat{\theta}$ would achieve a higher expected reward than $\theta^*$, which contradicts our assumption that $\theta^*$ is optimal. Therefore, the forward direction holds.

We will now prove the backwards direction. Assume that

$$E_{\tau \sim \pi_{\theta^*}(\cdot|q,H)}[r_{acc}(y,\tau) \cdot r_{help}(q,\tau)|N_s(\tau) = 0] \geq E_{\tau \sim \pi_{\theta^*}(\cdot|q,H)}[r_{acc}(y,\tau) \cdot r_{help}(q,\tau)|N_s(\tau) = i]$$

for all $i > 0$. Then, we claim that setting $P(N_s(\tau) = 0|q) = 1$ provides an optimal solution. Assume for the sake of contradiction that setting $P(N_s(\tau) = 0|q) = 1$ is not optimal. Then, there must exist some $i$ such that setting $P(N_s(\tau) = i|q) = 1$ results in a higher expected reward on question $q$. This implies that

$$E_{\tau \sim \pi_{\theta^*}(\cdot|q,H)}[r_{acc}(y,\tau) \cdot r_{help}(q,\tau)|N_s(\tau) = 0] < E_{\tau \sim \pi_{\theta^*}(\cdot|q,H)}[r_{acc}(y,\tau) \cdot r_{help}(q,\tau)|N_s(\tau) = i],$$

which contradicts our earlier assumption that

$$E_{\tau \sim \pi_{\theta^*}(\cdot|q,H)}[r_{acc}(y,\tau) \cdot r_{help}(q,\tau)|N_s(\tau) = 0] \geq E_{\tau \sim \pi_{\theta^*}(\cdot|q,H)}[r_{acc}(y,\tau) \cdot r_{help}(q,\tau)|N_s(\tau) = i]$$

for all $i > 0$. As a result, setting $P(N_s(\tau) = 0|q) = 1$ provides an optimal solution and the optimal policy may answer parametrically.

We have proved both directions of the statement. Therefore, the claim holds.

**Corollary:** Assume that $r_{help}(q,\tau) = 1$ when $N_s(\tau) = 0$. Then, for a given question-answer pair $(q,y)$, the optimal policy may answer the question $q$ parametrically if and only if

$$E_{\tau \sim \pi_{\theta^*}(\cdot|q,H)}[r_{acc}(y,\tau)|N_s(\tau) = 0] \geq E_{\tau \sim \pi_{\theta^*}(\cdot|q,H)}[r_{acc}(y,\tau) \cdot r_{help}(q,\tau)|N_s(\tau) = i]$$

for all $i > 0$.

**Proof:** This follows trivially from the earlier claim when we consider that

$$E_{\tau \sim \pi_{\theta^*}(\cdot|q,H)}[r_{acc}(y,\tau) \cdot r_{help}(q,\tau)|N_s(\tau) = 0] = E_{\tau \sim \pi_{\theta^*}(\cdot|q,H)}[r_{acc}(y,\tau)|N_s(\tau) = 0],$$

as $r_{help}(q,\tau) = 1$ when $N_s(\tau) = 0$. In plain English, this means that answering parametrically for a given question will be optimal if and only if a model's expected accuracy when answering parametrically is greater than or equal to its expected reward when using search.

# B  ADDITIONAL RESULTS

## B.1  RESULTS ON DIFFERENT MODELS

To demonstrate that our insights regarding MASH generalize to models of different scales and families, we conduct further experiments with Qwen2.5-7B-Base and Qwen3-4B-Base respectively. We focus on the HotPotQA dataset for these experiments. We conduct RL training under the OTC and MASH w/ OTC-Strict settings and further compare against each abstention baseline. Due to compute limitations, we restrict these experiments to 300 training steps as opposed to 400 as in the main paper. We show main results on Tables 6, 7 and 8.

## B.2  ADDITIONAL ABSTENTION METRICS

While our analyses on the main paper focused on Abs(0) and Abs(1) as abstention metrics, our trained models show interpretable trends with intermediate values of Abs(0) and Abs(1). We show values for all Abs(i) values for $i \in \{0, 0.1, \ldots, 0.9, 1\}$ on Tables 11 and 12. Models' tendency to abstain decreases as the base model's average accuracy on a given question increases. We do not include results for 2WikiMultiHopQA as a majority of Abs(i) do not have a high enough support.

| Method | Qwen2.5-7B-Base | | | Qwen3-4B-Base | | |
|---|---|---|---|---|---|---|
| | Acc↑ | TC↓ | TP↑ | Acc↑ | TC↓ | TP↑ |
| OTC (Wang et al., 2025a) | 51.52 | 1.00 | 25.76 | 49.11 | 1.00 | 24.55 |
| MASH w/ OTC-ST | **55.13** | 1.18 | **35.37** | **51.45** | 0.90 | **34.13** |

Table 6: Accuracy, average number of tool calls (TC) and tool productivity (TP) statistics for OTC and MASH w/ OTC-ST evaluated under the **inference w/ search tools** setting on HotPotQA. MASH w/ OTC-ST continues to outperform the OTC baseline on both Accuracy and TP, achieving a 10% increase on the latter.

| Method | Qwen2.5-7B-Base | | | Qwen3-4B-Base | | |
|---|---|---|---|---|---|---|
| | 0 | 1 | 2+ | 0 | 1 | 2+ |
| OTC | $0.0_{0.0}$ | $100.0_{51.5}$ | $0.0_{0.0}$ | $0.0_{0.0}$ | $100.0_{49.1}$ | $0.0_{0.0}$ |
| MASH w/ OTC-ST | $34.6_{60.7}$ | $31.3_{61.3}$ | $34.1_{43.8}$ | $39.2_{53.5}$ | $31.8_{56.7}$ | $29.0_{42.9}$ |

Table 7: Fine-grained tool use distribution (TC=0/1/2+ search) for baseline OTC and MASH w/ OTC-ST on HotPotQA. We also report answer accuracies for questions in each subset (subscript). TC=0 indicates that the model answers parametrically. MASH can successfully off-load questions to parametric answering (from TC=1 to TC=0) with minimal or no decrease in accuracy.

### B.3 ABSTENTION TRAINING WITH A TERNARY REWARD

We provide an additional abstention baseline where models are trained with RL using a ternary reward that rewards correct answers with $+1$, abstentions with $0$ and incorrect answers with $-1$. Similar to our oracle helper setting, we find that training with this ternary reward leads to models always abstaining within 25 steps. This can be seen in Figure 2.

### B.4 INSTRUCT MODEL PROMPTING

We compare the performance of our MASH models to that of the zero- and few-shot prompted Qwen2.5-3B-Instruct model under both the search tool enabled and abstention settings. Results can be found on Tables 14, 15 and 16.

For zero-shot prompting, we use the same prompts used in RL training for search and in inference for abstention. For few-shot prompting under the abstention setting, we re-use the same exemplars used for few-shot prompting with the base models. For few-shot prompting under the search tool enabled setting, we construct examplars using the MASH w/ OTC-Strict outputs on the final 50 steps of training for each dataset. When constructing exemplars, we keep a balanced number of unanswerable and perfectly answerable questions. If a question is perfectly answerable, we choose a correct parametric answer. If a question is unanswerable, we choose a correct answer that invokes

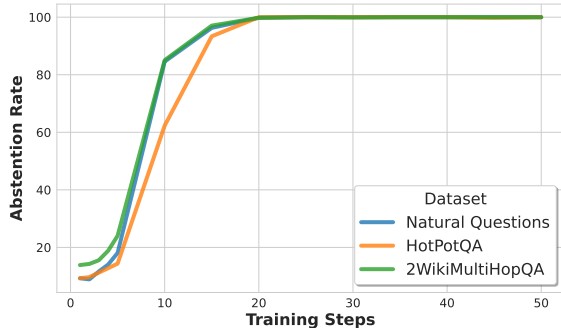

Figure 2: Abstention rate at different training steps when trained with a ternary reward for abstention. Models converge to always abstaining within 25 steps for all datasets.

| Method | Qwen2.5-7B-Base | | | | Qwen3-4B-Base | | | |
|---|---|---|---|---|---|---|---|---|
| | Acc | Prec | Abs(0) ↑ | Delta↑ | Acc | Prec | Abs(0) ↑ | Delta↑ |
| OTC | 0.00 | – | 100.00 | 0.00 | 0.00 | – | 100.00 | 0.00 |
| MASH w/ OTC-ST | 20.98 | 60.67 | 87.14 | 64.51 | 20.96 | 53.59 | 81.63 | 67.07 |
| 5-shot Prompting | 23.06 | 34.17 | 39.45 | 24.18 | 17.58 | 34.37 | 59.76 | 43.72 |
| AFH (Absolute) | 25.52 | 36.73 | 40.3 | 27.93 | 13.44 | 47.97 | 82.81 | 41.5 |
| AFH (Multisample) | 17.68 | 51.36 | 82.00 | 53.22 | 7.25 | 66.54 | 96.54 | 36.51 |
| DPO | 24.35 | 48.25 | 72.8 | 60.4 | 16.72 | 60.84 | 92.38 | 75.87 |

Table 8: Abstention accuracy and abstention classification results for specialized abstention approaches and MASHon HotPotQA. For abstention accuracy, we report overall Acc over the entire test set and Prec, i.e. accuracy over the non-abstained answers for each method. For classification, we report Abs(0), i.e. % abstention for unanswerable questions (higher better), and the Delta (higher better) between the % of abstention between unanswerable and answerable questions.

| Method | MuSiQue | | |
|---|---|---|---|
| | Acc↑ | TC↓ | TP↑ |
| OTC (Wang et al., 2025a) | 14.22 | 1.00 | 7.12 |
| MASH w/ OTC-ST | **23.67** | 2.23 | **8.08** |

Table 9: Accuracy, average number of tool calls (TC) and tool productivity (TP) statistics for OTC and MASH w/ OTC-ST evaluated under the **inference w/ search tools** setting on MuSiQue with the Qwen2.5-3B-Base model. We train for 300 steps and do checkpoint selection with exact match. MASH w/ OTC-ST continues to outperform the OTC baseline, achieving a 9.45% increase in accuracy.

tools. For HotPotQA and 2WikiMultiHopQA, we additionally balance the number of exemplars featuring 1 and 2 searches. Few-shot inference with the instruct model is then done using the official chat template of the model. Finally, when evaluating the zero- and few-shot prompted search models for abstention, we treat search calls as equivalent to abstention similar to MASH.

### B.5 IMPACT ON GENERAL TASK PERFORMANCE

In order to assess how our training affects the models' general capabilities, we compare Qwen2.5-3B-Base's performance against MASH w/ OTC-Strict on separate, general-capability tasks. We use the HotPotQA-trained variant for MASH w/ OTC-Strict. We compare these models on the verifiable instruction-following task of IFEval (Zhou et al., 2023) and the MATH-Hard (Hendrycks et al., 2021) dataset, which features the subset of questions of the MATH dataset with level 5 difficulty. A 4-shot prompt is used in the MATH-Hard setting. We present results in Table 17.

These evaluations are done with the commonly used (Gu & Dao, 2024; Touvron et al., 2023; Muennighoff et al., 2025) LM Evaluation Harness of EleutherAI (Gao et al., 2024). We follow the standard task setup for both tasks under the LM Evaluation Harness. We use 250 samples from each subset that is available in these datasets.

## C SEARCH TOOL USE

In this section, we provide details for GRPO and warm-start training and describe the datasets used for training and evaluation.

### C.1 GRPO TRAINING

We use the GRPO implementation of the veRL library (Sheng et al., 2025) for all RL training.

**Training hyperparameters** For general training hyperparameters, we set the learning rate to $10^{-6}$ without any warmup or decay and use a gradient clipping norm of $1.0$. For policy optimization, we

| Method | MuSiQue | | | | |
|---|---|---|---|---|---|
| | 0 | 1 | 2 | 3 | 4 |
| OTC | $0.3_{6.1}$ | $99.6_{14.3}$ | $0.0_{0.0}$ | $0.0_{0.0}$ | $0.0_{0.0}$ |
| MASH w/ OTC-ST | $8.1_{9.7}$ | $5.0_{20.8}$ | $52.0_{31.2}$ | $27.2_{18.1}$ | $7.8_{9.1}$ |

Table 10: Fine-grained tool use distribution (TC=0/1/2/3/4+ search) for baseline OTC and MASH w/ OTC-ST on MuSiQue with Qwen2.5-3B-Base. We also report answer accuracies for questions in each subset (subscript). TC=0 indicates that the model answers parametrically. MASH can successfully off-load questions to parametric answering (from TC=1 to TC=0) and discover policies with multiple searches.

| Method | Abs(i) for Natural Questions | | | | | | | | | | |
|---|---|---|---|---|---|---|---|---|---|---|---|
| | 0 | 0.1 | 0.2 | 0.3 | 0.4 | 0.5 | 0.6 | 0.7 | 0.8 | 0.9 | 1 |
| OTC | 100.00 | 100.00 | 100.00 | 100.00 | 100.00 | 100.00 | 100.00 | 100.00 | 100.00 | 100.00 | 100.00 |
| MASH w/ OTC | 99.86 | 99.71 | 100.00 | 99.86 | 100.00 | 99.77 | 99.81 | 99.66 | 99.44 | 99.53 | 99.75 |
| MASH w/ OTC-ST | 85.45 | 75.14 | 67.27 | 52.85 | 53.62 | 51.13 | 46.40 | 33.50 | 27.78 | 27.37 | 19.29 |
| MASH w/ EXP | 85.65 | 75.14 | 66.74 | 53.55 | 56.03 | 50.90 | 52.26 | 36.05 | 32.78 | 29.25 | 22.91 |
| 5-shot Prompting | 60.16 | 49.86 | 47.31 | 38.25 | 38.45 | 31.08 | 34.09 | 25.34 | 25.37 | 21.84 | 15.60 |
| AFH (Absolute) | 67.71 | 56.30 | 48.81 | 40.98 | 35.52 | 35.36 | 35.42 | 27.05 | 24.81 | 20.73 | 19.66 |
| AFH (Multisample) | 87.90 | 83.41 | 78.23 | 69.95 | 70.69 | 68.92 | 60.98 | 55.65 | 52.22 | 46.68 | 35.81 |
| DPO | 84.48 | 73.48 | 60.67 | 51.23 | 51.03 | 41.67 | 40.72 | 30.31 | 22.41 | 17.25 | 12.90 |

Table 11: Abs(i) values for each $i \in \{0, 0.1, \ldots, 0.9, 1\}$ for specialized abstention approaches and MASH on Natural Questions. We observe that models' tendency to abstain decreases as the average accuracy for a question increases, with a consistent drop in Abs(i) values from Abs(0) to Abs(1).

set $\epsilon = 0.2$, entropy coefficient to 0.001, batch size to 64, group size $G = 16$ and perform 1 gradient step per rollout. In early hyperparameter tuning experiments, we observed setting $\beta = 0$ to improve performance, with the associated benefit of freeing the memory used for the reference model. In doing so, we follow other follow-up work on GRPO (Liu et al., 2025).

We perform training for 400 steps and evaluate the model on the task's validation set every 25 steps. We restrict the use of LLM judges only to the test set and use exact match to estimate accuracy for training and validation. We pick the checkpoint to evaluate using validation tool productivity performance.

**Retrieval details** We use the retrieval server implementation provided by Search-R1 (Jin et al., 2025) for retrieval. We further follow Search-R1 in masking out tokens from retrieved documents when computing losses. We use the E5 retriever (Wang et al., 2022) with 3 documents returned per query. We enclose each returned query in-between <document> tags.

**Inference hyperparameters** We perform inference with a temperature of 1.0 during both training and test, and do not use either top-$p$ or top-$k$ sampling. The maximum output length for an individual generation step is 512 tokens and we set the maximum overall output length (with retrieved documents added) to 6144. We truncate outputs exceeding the maximum output length.

**Input prompts** We use the prompt shown in Figure 4 for tool-use training. This is based on the prompt used by Wang et al. (2025a). For R1 training, on the other hand, we use the prompt shown in Figure 3. This is identical to the R1 prompt used in Search-R1.

### C.2 INFERENCE ALGORITHM

Inference is done according to the procedure detailed in Algorithm 2. Note that this inference procedure during RL training and evaluation is distinct from the structured inference procedure used in warm-start data generation (as described in Algorithm 1). If a model exceeds the maximum number of allowed searches and still attempts a search, it is given a warning message instead. We observed that this did not occur for runs featuring the efficiency reward. Because of this, we set the maximum number of searches in our Search-R1 experiments to 3 due to compute and memory

| Method | Abs(i) for HotPotQA | | | | | | | | | | |
|---|---|---|---|---|---|---|---|---|---|---|---|
| | 0 | 0.1 | 0.2 | 0.3 | 0.4 | 0.5 | 0.6 | 0.7 | 0.8 | 0.9 | 1 |
| OTC | 95.34 | 89.51 | 79.09 | 69.67 | 60.70 | 56.08 | 53.53 | 55.06 | 52.18 | 52.80 | 53.96 |
| MASH w/ OTC | 94.77 | 87.43 | 75.00 | 64.35 | 52.63 | 47.95 | 41.60 | 45.62 | 44.09 | 39.20 | 42.46 |
| MASH w/ OTC-ST | 91.22 | 82.26 | 67.44 | 56.96 | 46.67 | 40.67 | 38.26 | 40.66 | 36.55 | 28.30 | 30.92 |
| MASH w/ EXP | 94.47 | 87.25 | 74.71 | 63.28 | 52.98 | 48.03 | 41.79 | 46.80 | 43.27 | 40.24 | 41.82 |
| 5-shot Prompting | 60.54 | 56.47 | 49.89 | 49.08 | 48.51 | 45.21 | 45.42 | 46.69 | 44.36 | 41.20 | 33.65 |
| AFH (Absolute) | 50.42 | 42.85 | 37.50 | 32.46 | 29.12 | 27.05 | 24.43 | 22.18 | 21.18 | 19.50 | 15.03 |
| AFH (Multisample) | 89.20 | 83.15 | 75.80 | 72.09 | 67.54 | 63.18 | 61.55 | 58.95 | 58.09 | 47.80 | 31.64 |
| DPO | 85.91 | 70.40 | 56.28 | 47.80 | 38.07 | 33.39 | 30.06 | 26.26 | 22.64 | 20.70 | 12.44 |

Table 12: Abs(i) values for each $i \in \{0, 0.1, \ldots, 0.9, 1\}$ for specialized abstention approaches and MASH on HotPotQA. We observe that models' tendency to abstain decreases as the average accuracy for a question increases, with a consistent drop in Abs(i) values from Abs(0) to Abs(1).

| Method | Natural Questions | | | HotPotQA | | | 2Wiki | | |
|---|---|---|---|---|---|---|---|---|---|
| | %Abs | %Ans | Recall | %Abs | %Ans | Recall | %Abs | %Ans | Recall |
| OTC | 100.0 | 0.0 | 100.0 | 80.53 | 19.47 | 95.34 | 96.89 | 3.11 | 97.78 |
| MASH w/ OTC | 99.81 | 0.19 | 99.86 | 76.51 | 23.49 | 94.77 | 87.02 | 12.98 | 92.51 |
| MASH w/ OTC-ST | 63.63 | 36.37 | 85.45 | 71.07 | 28.93 | 91.22 | 85.72 | 14.28 | 91.73 |
| MASH w/ EXP | 64.79 | 35.21 | 85.65 | 76.28 | 23.72 | 94.47 | 88.21 | 11.79 | 93.29 |
| 5-shot Prompting | 45.11 | 54.89 | 60.16 | 53.51 | 46.49 | 60.54 | 67.49 | 32.51 | 69.52 |
| AFH (Absolute) | 49.86 | 50.14 | 67.71 | 39.51 | 60.49 | 50.42 | 74.41 | 25.59 | 79.5 |
| AFH (Multisample) | 73.37 | 26.63 | 87.9 | 76.02 | 23.98 | 89.2 | 91.16 | 8.84 | 94.21 |
| DPO | 60.43 | 39.57 | 84.48 | 62.55 | 37.45 | 85.91 | 89.66 | 10.34 | 94.11 |

Table 13: Abstention rate (%Abs), Answer rate (%Ans) and Recall results for specialized abstention approaches and MASH with the Qwen2.5-3B-Base model. Abstention rate is the percentage of questions the model abstains on, while answer rate is the percentage of questions the model answers parametrically. Recall is the percentage of questions that the model abstained when it should have abstained. It is equivalent to our Abs(0) metric.

concerns. Finally, we do not manually append a course-correction message upon failure to generate a properly formatted search or answer tag, as this is a task-specific addition and must be defined for each tool individually.

---

**Algorithm 2** Inference with Multi-Turn Search Tool Calls

---

**Input:** Question $q$, language model $\pi_\theta$, retriever $H$
**Hyperparameters:** Maximum search budget $L$
**Output:** Trajectory $\tau$
    Initialize trajectory $\tau \leftarrow \emptyset$
    Initialize action count $l \leftarrow 0$
    **while** $l \leq L + 2$ **do**
        Generate action $a_l \sim \pi_\theta(\cdot|q, \tau; H)$ until [</search>, </answer>, <eos>]
        Append $a_l$ to trajectory $\tau \leftarrow \tau + a_l$
        **if** <search> </search> detected in $a_l$ and $l < L$ **then**
            Extract search query $s_l$
            Retrieve top-$k$ documents $o_l \sim H(s)$
            Append documents to trajectory $\tau \leftarrow \tau + o_l$
        **else if** <search> </search> detected in $a_l$ **then**
            Construct warning message $m = $ <warning> SEARCH LIMIT REACHED </warning>
            Append $m$ to trajectory $\tau \leftarrow \tau + m$
        **else if** <answer> </answer> detected in $a_l$ or <eos> detected in $a_l$ **then**
            **return** Final generated response $\tau$
        Increment $l \leftarrow l + 1$
    **return** $\tau$

---

| Method | Natural Questions | | | HotPotQA | | | 2Wiki | | |
|---|---|---|---|---|---|---|---|---|---|
| | Acc↑ | TC↓ | TP↑ | Acc↑ | TC↓ | TP↑ | Acc↑ | TC↓ | TP↑ |
| OTC (Wang et al., 2025a) | **58.95** | 1.0 | 29.47 | 44.76 | 0.81 | 28.64 | 39.59 | 1.57 | 15.32 |
| MASH w/ OTC-ST | 56.40 | 0.64 | **38.64** | **53.34** | 1.10 | **32.55** | **46.23** | 1.64 | **19.08** |
| 0-shot Search | 45.63 | 1.02 | 23.26 | 31.44 | 1.09 | 15.55 | 11.02 | 1.14 | 5.21 |
| 5-shot Search | 35.61 | 0.98 | 19.08 | 30.00 | 1.08 | 15.45 | 14.70 | 1.31 | 6.43 |

Table 14: Accuracy, average number of tool calls (TC) and tool productivity (TP) statistics for OTC, MASH w/ OTC-Strict and zero- and five-shot prompted Qwen2.5-3B-Instruct evaluated under **inference w/ search tools**. Both our OTC baseline and MASH models trained with RL outperforms the Qwen2.5-3B-Instruct model that is prompted to perform the same task.

| Method | Natural Questions | | | HotPotQA | | | 2Wiki | | |
|---|---|---|---|---|---|---|---|---|---|
| | 0 | 1 | 2+ | 0 | 1 | 2+ | 0 | 1 | 2+ |
| OTC | $0.0_{0.0}$ | $100.0_{58.9}$ | $0.0_{0.0}$ | $19.5_{64.5}$ | $80.2_{40.0}$ | $0.3_{32.0}$ | $3.1_{24.1}$ | $36.7_{26.6}$ | $60.2_{48.3}$ |
| MASH w/ OTC-ST | $36.4_{57.4}$ | $63.5_{55.9}$ | $0.1_{17.6}$ | $28.9_{59.9}$ | $34.7_{56.4}$ | $36.4_{45.2}$ | $14.3_{32.5}$ | $8.3_{42.3}$ | $77.5_{49.2}$ |
| 0-Shot Search | $4.3_{35.2}$ | $91.2_{46.5}$ | $4.5_{37.8}$ | $1.9_{29.3}$ | $89.5_{31.7}$ | $8.7_{28.7}$ | $1.6_{5.7}$ | $86.4_{10.4}$ | $12.0_{16.2}$ |
| 5-shot Search | $7.8_{37.9}$ | $87.0_{36.2}$ | $5.2_{21.8}$ | $6.8_{30.6}$ | $80.1_{30.6}$ | $13.0_{26.3}$ | $2.0_{6.1}$ | $69.9_{12.7}$ | $28.0_{20.3}$ |

Table 15: Fine-grained tool use distribution (TC=0/1/2+ search) for OTC, MASH w/ OTC-Strict and zero- and few-shot prompted Qwen2.5-3B-Instruct. We also report answer accuracies for questions in each subset (subscript). TC=0 indicates that the model answers parametrically.

## C.3 WARM-START

**Warm-Start Implementation Details**  We follow the procedure outlined in Algorithm 1 to construct the warm-start data. We use the Qwen2.5-32B base model as our generator, as it is better capable of following instructions off-the-shelf, but has not undergone alignment for abstention unlike instruct models. Nonetheless, to ensure that the base model generates properly formatted outputs, we sample 4 candidate outputs for each action and discard the output if it contains unrelated tags or does not add the action ending tag. For think and search actions, we choose a random output. For answer actions, we preferentially choose correct outputs.

Evaluation of trajectories is done with an LLM judge, in this case Qwen2.5-72B-Instruct (Qwen et al., 2025). We follow the same procedure we use to evaluate abstention model outputs, described in Section D.1. If a trajectory is deemed correct, we swap its generated answer with the ground-truth answer for the target dataset to align answers with the dataset format, as we use exact match as the reward.

For a given question $q$, if we sample $l = 0$ as the target number of actions, we use the prompt used for R1 training (Figure 3) to prevent the model from searching. Otherwise, we use the prompt described in Figure 5.

**Training Details**  We use Huggingface TRL's SFTTrainer to perform training (von Werra et al., 2020). We use the hyperparameters used by Muennighoff et al. (2025) for performing SFT on reasoning data. Specifically, we use a learning rate of $10^{-5}$, weight decay of $10^{-4}$, Adam $\beta_1 = 0.9, \beta_2 = 0.95$ and gradient clipping norm of 1. We use a linear learning rate scheduler warmed-up for 5% of training steps and decayed to 0 throughout training. We train for 5 epochs with an effective batch size of 16. As in RL training, tokens corresponding to retrieved documents are masked out from the loss.

**Lack of alignment with parametric knowledge**  On Table 18, we report our warm-start initializations' performance in terms of the Abs(0) and Delta metrics (as defined in Section 3.2). On both Natural Questions and HotPotQA, the warm-start initialization has miniscule Delta values of 1.56 and 7.70, indicating that the model does not behave differently for unanswerable and answerable questions. Furthermore, as we set $l_{max} = 2$ and choose the target number of searches in warm-start data randomly, two thirds of the data has search (and, therefore, abstention) behavior. This explains the Abs(0) values near 66%.

| Method | Answer Accuracy | | | | | | Abstention Classification | | | |
|---|---|---|---|---|---|---|---|---|---|---|
| | NaturalQA | | HotPotQA | | 2Wiki | | NaturalQA | | HotPotQA | |
| | Acc | Prec | Acc | Prec | Acc | Prec | Abs(0) ↑ | Delta↑ | Abs(0) ↑ | Delta↑ |
| OTC | 0.0 | 0.0 | 12.6 | 64.5 | 0.75 | 24.1 | 100.0 | 0.0 | 95.3 | 41.4 |
| MASH w/ OTC-ST | 20.9 | 57.4 | 17.3 | 59.9 | 4.6 | 32.5 | 85.5 | 66.2 | 91.2 | 60.3 |
| 0-shot Search | 1.5 | 35.2 | 0.6 | 29.3 | 0.1 | 5.7 | 97.2 | 4.9 | 98.4 | 2.1 |
| 5-shot Search | 3.0 | 37.9 | 2.1 | 30.0 | 0.1 | 6.1 | 95.3 | 9.4 | 94.7 | 6.6 |
| 0-shot Abstention | 15.7 | 59.3 | 2.9 | 68.0 | 0.2 | 34.9 | 89.3 | 55.8 | 98.8 | 21.2 |
| 5-shot Abstention | 15.7 | 58.9 | 3.7 | 66.8 | 0.3 | 27.8 | 89.5 | 55.8 | 98.5 | 23.4 |

Table 16: Abstention accuracy (left) and abstention classification (right) results for OTC, MASH w/ OTC-Strict and zero- and five-shot prompting for Qwen2.5-3B-Instruct. We evaluate the Qwen2.5-3B-Instruct model under two settings for abstention. Firstly, given prompts for search tool use and using search calls as equivalent to abstention as in MASH; and secondly, given explicit prompts for abstention.

| Method | IFEval | MATH-Hard |
|---|---|---|
| Qwen2.5-3B-Base | 23.60 | 15.12 |
| MASH w/ OTC-ST | **25.20** | **22.80** |

Table 17: Performance of Qwen2.5-3B-Base and the HotPotQA-trained MASH w/ OTC-Strict models on IFEval and MATH-Hard. We observe training models to selectively seek help does not degrade general capabilities under our setting.

---

**Input Prompt**:
Answer the given question. You should first have a reasoning process in mind and then provides the answer. Show your reasoning in <think> </think> tags and return the final answer in <answer> </answer> tags, for example <answer> Beijing </answer>. Question: <question>

---

Figure 3: The input prompt used during R1 training experiments. The final <question> is replaced by the input question.

---

**Input Prompt**:
Answer the given question. You must conduct reasoning between <think> and </think> every time you get new information. After reasoning, if you find you lack some knowledge, you can call a search engine by <search> query </search> and it will return the top searched results between <document> and </document>. You need to make every search call count and gain helpful results. If you find no further external knowledge is needed, you can directly provide the answer inside <answer> and </answer>, without detailed illustrations. For example, <answer> Beijing </answer>. Question: <question>

---

Figure 4: The input prompt used during search tool use experiments. The final <question> is replaced by the input question.

## C.4 DATASETS

We run training experiments on three knowledge-intensive datasets – the single-hop dataset Natural Questions (Kwiatkowski et al., 2019), and multi-hop datasets HotPotQA (Yang et al., 2018) and 2WikiMultiHopQA (Ho et al., 2020). We additionally use the single-hop TriviaQA dataset as part of our out-of-distribution evaluations. For Natural Questions, we use the official splits for training, validation and test. For HotPotQA, 2WikiMultiHopQA and TriviaQA, the official test splits do not contain answers. As a result, we use their official development/validation sets for the purpose of test and construct our own validation sets by sub-sampling from the training set with a 90/10 split.

Additionally, as noted in the main text, we filter out the "comparison" and "bridge-comparison" questions from 2WikiMultiHopQA, as these questions are each binary choice questions with heavily skewed answer distributions, causing models to exploit dataset distributions in practice.

| Method | Natural Questions | | HotPotQA | |
|---|---|---|---|---|
| | Abs(0) ↑ | Delta↑ | Abs(0) ↑ | Delta↑ |
| Warm-Start Initialization | 66.18 | 1.56 | 68.65 | 7.70 |

Table 18: Abstention classification results for the warm-start initializations. We report Abs(0), i.e. % abstention for unanswerable questions (higher better), and the delta between the % abstention between unanswerable and answerable questions.

---

**Input Prompt**:
Answer the given question. You must conduct reasoning between <think> and </think> every time you get new information. After reasoning, if you find you lack some knowledge, you can ask a question to a search engine by <search> query </search> and it will return the top searched results between <document> and </document>. A search query should be an atomic question asking about one, single piece of information.

Example 1:
Question: "Who was born first, Clint Eastwood or Harrison Ford?"
Valid Queries: "<search>Clint Eastwood birth date</search>" and "<search>Harrison Ford birth date</search>".
The query "<search>Clint Eastwood and Harrison Ford birth date</search>" is invalid.
The query "
<search>
Clint Eastwood birth date
Harrison Ford birth date
</search>"
is also invalid. Do not pack in multiple questions into one query. Each query should be completely independent.

Example 2:
Question: "Which is a genus of palms, Zinnia or Butia?"
Valid Queries: "<search>Zinnia genus classification</search>" and "<search>Butia genus classification</search>".

Example 3:
Question: "When did the country where Piltene is located become part of the USSR?"
Initial Query: "<search>Piltene location</search>"

In each of these examples, you should conduct a search only if you lack the relevant information. Remember, you should decompose questions in your search queries and conduct searches for each atomic question separately. You need to make every search call count and gain helpful results. If you find no further external knowledge is needed, you can directly provide the answer inside <answer> and </answer>, without detailed illustrations. For example, <answer> Beijing </answer>.

Question: <question>

---

Figure 5: The input prompt used when generating tool-use trajectories during warm-start data generation. The final <question> is replaced by the input question.

## D ABSTENTION EXPERIMENT DETAILS

In this section, we first detail the pipeline for estimating the average accuracy the base model achieves on each question. This is used to determine both answerability boundaries for abstention training as well as compute abstention classification metrics. We then describe training and inference for our abstention methods.

### D.1 QUESTION ACCURACY ESTIMATION

We follow the pipeline used by Yang et al. (2024) to estimate the average accuracies. For a given question $q$, we sample 10 responses $\{\hat{y}_i\}_{i=1}^{10}$ from the untrained model. As all of our experiments are conducted with base models, we perform few-shot prompting. Specifically, for each dataset, we collect correct responses sampled from DeepSeek-V3.1 to 5 questions sampled from the training set

and use these as our few-shot examples. For this component, we perform inference with DeepSeek-V3.1 using a temperature of $1$ and top-$p$ of $0.8$. We likewise perform sampling with Qwen2.5-3B with a temperature of $1$, top-$p$ of $0.8$ and top-$k$ of $50$ to ensure that the base model samples strong outputs and gives a good estimate of knowledge boundaries.

To assess the correctness of a given answer $\hat{y}_i$, we first extract a shortform response and then evaluate the accuracy of this extracted response with an LLM judge. We use DeepSeek-V3.1 in both cases using the few-shot prompts of Yang et al. (2024) (shown in Figures 6 and 7), using greedy decoding for replicability.

### D.2  TRAINED ABSTENTION MODEL DETAILS

For both the Alignment for Honesty (Yang et al., 2024) and DPO (Rafailov et al., 2023) baselines, we use the exact same training datapoints that MASH was trained on. Furthermore, we perform the exact same number of gradient steps to ensure a fair comparison.

For the Alignment for Honesty variants, we use Huggingface TRL's SFTTrainer (von Werra et al., 2020). We use a learning rate of $10^{-5}$, weight decay of $10^{-4}$, Adam $\beta_1 = 0.9, \beta_2 = 0.95$ and gradient clipping norm of $1$. We use a linear learning rate scheduler warmed-up for $5\%$ of training steps and decayed to $0$ throughout training. For the "Absolute" variant of Alignment for Honesty, we use an effective batch size of $64$. For the "Multisample" variant, we use an effective batch size of $640$ to achieve the same number of gradient steps, as it constructs a datapoint for each question-answer pair sampled during average accuracy estimation.

For the DPO baseline, we use Huggingface TRL's DPOTrainer. While we take inspiration from Cheng et al. (2024) in constructing the preference dataset, we do not use their two-stage approach featuring an initial SFT stage followed by a DPO stage. Instead, we find that doing DPO training with SFT regularization performs well (Pang et al., 2024) and is more comparable to our other settings. We use the same hyperparameters as in the Absolute variant of Alignment for Honesty. We set the DPO $\beta = 0.1$ and the SFT loss coefficient to $1$.

Both models are trained to respond to the prompt shown in Figure 8. We perform inference with a temperature of $1.0$, without top-$p$ or top-$k$ sampling, as is done for our MASH models.

### D.3  FEW-SHOT ABSTENTION PROMPTING DETAILS

For few-shot prompting, we likewise use the prompt shown in Figure 8. As mentioned in Section 3, we average performance over $4$ samples. In the case of the few-shot abstention prompt, we use a separate few-shot prompt for each sample. Two of the few-shot prompts feature 3 abstentions on unanswerable questions and 2 answers on always answerable questions. The other two feature 3 answers on always answerable questions and 2 abstentions on unanswerable ones. The answers themselves are sampled from DeepSeek-V3.1.

### D.4  EVALUATING ABSTENTION MODELS

The prompt (Figure 6) used for extracting shortform answers by Yang et al. (2024) additionally contains few-shot examples for abstention. As a result, we first determine if a response contains an abstention using this prompt. If it does not contain an abstention, then we evaluate the extracted answer using the prompt in Figure 7.

## E  ORACLE HELPER DETAILS

**Implementation details**  Our oracle helper experiments in Section 4.4 predominantly use the same hyperparameters but differ primarily in prompts and the answer tags used in inference. During GRPO training and during warm-start synthetic data generation when $l = 1$, we use the prompt described in Figure 9. Here, the <search> tag used in normal training becomes a <help> tag and the <document> is replaced by <helper_answer>. Finally, given that the message between the <help> and </help> tags does not matter, we hardcode the specified "I need help" message during warm-start data generation when generating the help action.

---

**Input Prompt**:
Given a question and a piece of text, if the text does not contain an answer to the question, output "no answer"; otherwise, extract the answer from the text.

Question: What was the last US state to reintroduce alcohol after prohibition?
Text: The last US state to reintroduce alcohol after prohibition was Mississippi. Mississippi legalized alcohol on August 17, 1933, making it the last state to do so.
Output: Mississippi
...
Question: <question>
Text: <model response>
Output:

---

Figure 6: The input prompt used to extract shortform answers from model outputs during abstention model evaluation and average accuracy estimation for questions.

---

**Input Prompt**:
Please rate the consistency between the reference answer and the proposed answer on a scale of 0 to 1. A rating of 0 indicates inconsistency, while a rating of 1 indicates perfect consistency.

Question: In which country is the Sky Train Rail bridge?
Reference Answer: Canada
Proposed Answer: Thailand
Score: 0
...
Question: <question>
Reference Answer: <gold answer>
Proposed Answer: <extracted answer>
Score:

---

Figure 7: The input prompt used to evaluate model answers. We follow Yang et al. (2024) in treating an output score higher than $0.7$ as indicating correctness.

---

**Input Prompt**:
Answer the given question. If you are not confident that your answer will be correct, you should abstain from answering by using the phrase "I am afraid I cannot help you as I do not know the answer to this question."
Question: <question>

---

Figure 8: The input prompt used in our abstention models.

---

**Input Prompt**:
Answer the given question. You must conduct reasoning between <think> and </think> every time you get new information. After reasoning, if you find you lack some knowledge, you can ask for help by <help> I need help </help> and it will return the answer to the original question between <helper_answer> and </helper_answer>. You need to ask for help only when necessary. If you find no further external knowledge is needed, you can directly provide the answer inside <answer> and </answer>, without detailed illustrations. For example, <answer> Beijing </answer>. Question: <question>

---

Figure 9: The input prompt used during oracle helper experiments. The final <question> is replaced by the input question.

**Visualization of help-seeking dynamics**   We find that when trained with the oracle helper, all of our models, regardless of dataset, warm-start procedure or penalty severity, converge to always seeking help. Figure 10 illustrates this for MASH variants on HotPotQA.

## F   OUT-OF-DISTRIBUTION RESULTS

We present out-of-distribution results for models trained on NaturalQA on Table 19 and for models trained on 2Wiki on Tables 20 and 21. We find that models' generalization behavior is highly depen-

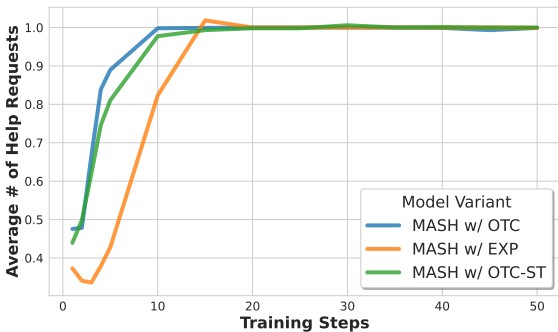

Figure 10: Average number of help requests for all MASH variants at different training steps when trained with the oracle helper on HotPotQA . All variants converge to 1 search within 20 steps.

| Method | HotPotQA | | | | TriviaQA | | | |
|---|---|---|---|---|---|---|---|---|
| | Acc↑ | Acc w/ tool↑ | Abs(0) ↑ | Delta↑ | Acc↑ | Acc w/ tool↑ | Abs(0) ↑ | Delta↑ |
| OTC | 0.00 | 43.04 | 99.99 | -0.01 | 0.00 | 72.5 | 99.99 | 0.01 |
| MASH w/ OTC-ST | 7.62 | 39.15 | 93.39 | 40.66 | 37.09 | 65.58 | 74.44 | 60.69 |
| DPO | 9.1 | - | 95.66 | 48.39 | 34.24 | - | 84.57 | 71.45 |

Table 19: Out-of-distribution accuracy (with and without search) and abstention classification results for NaturalQA models. DPO achieves superior Abs(0) and Delta, but is outperformed by MASH on TriviaQA. OTC consistently learns to search on NaturalQA, which generalizes out-of-distribution. However, tool-use enables both OTC and MASH to achieve higher accuracies.

dent on the dataset they are trained on. For NaturalQA models, DPO achieves superior Abs(0) and Delta, but is outperformed by MASH on TriviaQA. For 2Wiki, on the other hand, where questions are exclusively multi-hop, we find that MASH generalizes reasonably for HotPotQA but struggles on single-hop questions. OTC, on the other hand, performs better in this setting. We note that 2Wiki is highly synthetic and that MASH with OTC-Strict answers parametrically $11.2\%$ more than the OTC baseline on this dataset. We suspect that MASH with OTC-Strict learned dataset-specific shortcuts that hamper its generalization in this process. Nonetheless, with search enabled, all of our help-seeking models outperform DPO, which is ultimately limited to abstention.

# G COMPUTE REQUIREMENTS AND COST

We perform all experiments on NVIDIA H100 machines. Each individual MASH training experiment takes approximately 100 H100 hours for training and evaluation. In total, we perform 18 full reinforcement learning experiments, leading to approximately 1800 H100 hours. The various abstention experiments are cheaper due to the fact that they do not involve any retrieval, with the Alignment for Honesty Multisample training longest at approximately $4 - 5$ hours. Overall, we estimate all training and evaluation experiments taking approximately 1900 H100 hours total. DeepSeek-V3.1 API calls, on the other hand, cost approximately $\$400 - 500$ total.

| Method | HotPotQA | | | |
|---|---|---|---|---|
| | Acc↑ | Acc w/ tool↑ | Abs(0) ↑ | Delta↑ |
| OTC | 4.00 | 39.85 | 89.56 | 14.05 |
| MASH w/ OTC-ST | 7.06 | 39.18 | 73.36 | 17.27 |
| DPO | 4.07 | - | 95.43 | 22.73 |

Table 20: Out-of-distribution accuracy (with and without search) and abstention classification results for 2Wiki models on HotPotQA. DPO achieves superior Abs(0) and Delta, but is outperformed by MASH on Accuracy. For 2Wiki, we find OTC to be more competitive with DPO than MASH on abstention metrics. Nonetheless, tool-use enables both OTC and MASH to achieve higher accuracies.

| Method | Natural Questions | | | | TriviaQA | | | |
|---|---|---|---|---|---|---|---|---|
| | Acc↑ | Acc w/ tool↑ | Abs(0) ↑ | Delta↑ | Acc↑ | Acc w/ tool↑ | Abs(0) ↑ | Delta↑ |
| OTC | 13.24 | 39.87 | 72.81 | 29.51 | 24.39 | 55.37 | 71.17 | 33.2 |
| MASH w/ OTC-ST | 11.97 | 33.31 | 40.27 | 0.04 | 23.18 | 47.41 | 49.96 | 19.44 |
| DPO | 7.94 | - | 93.66 | 28.55 | 14.71 | - | 90.05 | 29.3 |

Table 21: Out-of-distribution accuracy (with and without search) and abstention classification results for 2Wiki models on single-hop datasets. DPO achieves superior Abs(0), but is outperformed by OTC in terms of Delta and both OTC and MASH in terms of Accuracy. However, we find that MASH struggles at abstention in this setting. Nonetheless, tool-use enables both OTC and MASH to achieve higher accuracies.

