# OpenReview forum: "Pay-Per-Search Models Are Abstention Models"
_ICLR.cc/2026/Conference — Submitted to ICLR 2026_

### Official Review · Reviewer_b1mW · 2025-10-31

**Soundness:** 3
**Presentation:** 3
**Contribution:** 2
**Rating:** 4
**Confidence:** 4

**Summary:**

This paper introduces MASH, a reinforcement learning framework that models abstention through selective help-seeking. During training, the model learns when to use external tools; at inference, removing the tools turns help-seeking into abstention, improving reliability and efficiency on QA tasks.

**Strengths:**

The paper presents a clear framework for learning abstention via selective help-seeking. Its design—training with retrieval tools and inferring without them—elegantly links tool-use efficiency to abstention behavior. Across multiple QA datasets, MASH delivers strong empirical gains, including higher tool productivity and a 7.6% accuracy improvement on multi-hop QA.

**Weaknesses:**

1. The method optimizes a proxy objective—binary correctness multiplied by a search penalty—while abstention is induced post hoc by removing tools at inference rather than being learned directly. The paper offers no theoretical account of why this proxy should produce a stable abstention boundary; the mechanism remains unclear and appears incidental.
2. In the oracle-helper setting, the optimal policy is trivially to always ask for help, since help deterministically returns the gold answer. A model will therefore query every time. This biased environment cannot substantiate the claim that “this setting with the oracle helper is equivalent to explicitly training for abstention using RL”; it conflates environmental bias with algorithmic behavior.
3. The approach appears tailored to QA with a specific retrieval setup, and its generalization beyond that scope is unproven. Across OOD and several QA datasets, the method does not show a clear advantage over DPO, which directly optimizes abstention behavior. This suggests that the observed gains may depend heavily on dataset characteristics rather than reflecting a genuinely more general or reliable abstention mechanism.

If my questions are resolved, I will consider raising the score.

**Questions:**

1. Why should the “correctness × search-penalty” proxy and constraining tool usage induce meaningful “abstention”? Provide a theoretical rationale or explicit assumptions
2. For each dataset, could you report answer rate, abstention rate, and recall to provide a more complete analysis of the model’s abstention behavior?
3. Wouldn’t it be more direct to train abstention explicitly with a ternary reward (e.g., Correct = +1, Abstain = 0, Wrong = −1), rather than relying on the binary correctness × search-penalty proxy? Could you try continuing RL from a Search-R1 checkpoint using this objective to see whether it leads to a clearer or more consistent abstention behavior?
4. To rule out possible dataset coincidence, could you expand the experiments to more QA datasets and evaluate them, to examine whether the model still learns a stable and interpretable abstention boundary across different distributions?

---

> ### Author Response · Authors · 2025-11-24
> **Official Response to Reviewer b1mW Part 1**
>
> **Response to Weakness 1**
>
> We thank the reviewer for their suggestion to place a theoretical analysis of our proxy objective. Its inclusion will improve the soundness of our paper.
>
> Let $\pi_{\theta^{\star}}(\tau \mid q, H)$
> be the probability of trajectory $\tau$ given question $q$ and helper $H$, and let $N_s(\tau)$ be the number of searches in $\tau$. Then the optimal policy will answer question $q$ parametrically if and only if $E_{\\tau \\sim \\pi_{\\theta^{\star}}}\\!\\left[r_{\\mathrm{acc}}(y,\\tau)*r_{\\mathrm{help}}(q,\\tau)\\mid N_s(\\tau)=0\\right] \\ge E_{\\tau \\sim \\pi_{\\theta^{\star}}}\\!\\left[r_{\\mathrm{acc}}(y,\\tau)*r_{\\mathrm{help}}(q,\\tau)\\mid N_s(\\tau)=i\\right]$  for all $i>0$.
>
>
>
>
> **We provide a formal proof of this in Section A of the Appendix.** To concretize this with an example, consider the oracle helper setting and let the help penalty be $r_{help}(y, \tau) = \lambda^{N_s(\tau)}$, where $0 < \lambda < 1$ is a hyperparameter controlling the severity of the penalty and $N_s(\tau)$ is the number of searches performed by $\tau$. Then, the model will answer parametrically if its average accuracy is greater than or equal to $\lambda$. Particularly, it will exclusively answer parametrically if its average accuracy is greater than $\lambda$.
>
> For regular helpers, this boundary will depend both on the strength of the retrieval penalty and on the reliability of the help. Therefore, during our RL training, the model simultaneously learns how to seek external help, and when to do so based on its parametric knowledge and its implicit estimate of the helper's reliability.
>
> Note that we make two trivial assumptions in our proof. First, we set $\beta=0$ for the KL penalty to focus solely on the behavior of the policy that maximizes reward. Secondly, we assume that the optimal policy cannot achieve perfect accuracy, i.e. answer all questions correctly parametrically with P(y | q) = 1. This is because the optimal policy cannot acquire new knowledge beyond the base model's knowledge boundaries during RL; it can only learn how to align its search behavior with its knowledge.

---

> ### Author Response · Authors · 2025-11-24
> **Official Response to Reviewer b1mW Part 2**
>
> **Response to Weakness 2 and Question 3**
>
> *“In the oracle-helper setting, the optimal policy is trivially to always ask for help, since help deterministically returns the gold answer. A model will therefore query every time.“*
>
> **Response:** This is incorrect. As we describe in the theoretical argument above, the optimal policy will answer parametrically when its average accuracy is greater than the expected reward obtained when using the oracle helper. Concretely, for a search penalty of, for instance, 0.7 for 1 help call, the optimal policy will only always ask for help if it cannot answer any question with an average accuracy greater than 0.7 in the entire dataset.
>
> *“This biased environment cannot substantiate the claim that “this setting with the oracle helper is equivalent to explicitly training for abstention using RL”; it conflates environmental bias with algorithmic behavior.”*
>
> *"Question 3: Wouldn’t it be more direct to train abstention explicitly with a ternary reward (e.g., Correct = +1, Abstain = 0, Wrong = −1), rather than relying on the binary correctness × search-penalty proxy?"*
>
> **Response:** Let us explain this clearly below, as we did not have the space in the paper to explain this clearly. But this is an important point that we will expand on in the next version of the paper.
>
> First, let us substantiate our claim with empirical results. We trained models for abstention directly with the suggested ternary reward formulation for all three datasets. For all settings, models converged to always abstaining within 25 training steps, which is similar to the behavior observed under the oracle helper setting. We include graphs for this in Appendix B.3.
>
> Let us now expand on our claim that the oracle helper setting and direct abstention setting are conceptually equivalent. We argue that the optimal policy for direct abstention training and the oracle helper setting are the same.
>
> First, remember that our oracle helper setting is a simplified version of the earlier experiments – the model only generates the help tag, i.e. no search query, and the oracle helper returns the final correct answer (not answers to a model-specified query). If help is invoked, the optimal policy can trivially learn to copy this as the final correct answer. Therefore, invoking the oracle helper will always lead to a correct answer, i.e. $r_{acc} = 1$ and therefore, $r = r_{help}$.
>
> Our main argument here is that, given a question $q$ and after having invoked one search, the optimal policy in the oracle helper setting, by definition, cannot generate an incorrect answer as it has access to the correct answer. For the same reason, it is not incentivized to invoke any additional searches even though it technically can.
>
> Therefore, the ternary reward structure of this oracle helper setting is exactly the same as that of direct abstention training – reward 0 for incorrect answers, reward 1 for correct answers, and $r_{help}$ for seeking help. If $r_{help}=0.5$, this will have the exact same hyperparameters as the above direct abstention training experiment, simply rescaled and shifted.
>
> *“Wouldn’t it be more direct to train abstention explicitly with a ternary reward (e.g., Correct = +1, Abstain = 0, Wrong = −1), rather than relying on the binary correctness × search-penalty proxy?”*
>
> **Response:** Furthermore, as we emphasize in our paper, MASH training allows the model to learn two tasks simultaneously. In addition to abstention, it learns how to search to answer questions it wouldn’t otherwise be able to solve. Training for abstention alone reduces harm by limiting erroneous outputs, but does not increase the number of questions the model could solve. MASH, we argue, elegantly combines the two objectives.

---

> > ### Author Response · Authors · 2025-11-24
> > **Official Response to Reviewer b1mW Part 3**
> >
> > **Response to Weakness 3 and Question 4**
> >
> > *"The approach appears tailored to QA with a specific retrieval setup, and its generalization beyond that scope is unproven."*
> >
> > We follow prior literature on abstention [1,2,3,4,5,6] and use short-form QA for evaluation. We assume that prior works make this choice because reliable evaluation of long-form answer correctness is still an open problem. While extending our framework on these other settings or other domains such as math or coding is interesting, and will likely invite unique challenges, we assert that this is beyond the scope of our work.
> >
> > *“Across OOD and several QA datasets, the method does not show a clear advantage over DPO, which directly optimizes abstention behavior.”*
> >
> > We would like to push back against the claim that MASH does not show a clear advantage over DPO. Training for abstention alone with DPO reduces harm by limiting erroneous outputs, but does not increase the number of questions the model could solve. Our proxy objective enables both learning an abstention boundary while also teaching the model the ability to seek help (for instance by using search), which it can then use to solve questions it otherwise wouldn’t be able to solve.
> >
> > Furthermore, note that we invariably favored DPO (and the SFT-based Alignment for Honesty baselines) during training. These required first determining the ground-truth knowledge-boundaries for which we use an LLM judge. In contrast, MASH uses the much noisier exact match reward during training, which often incorrectly penalizes correct model outputs, due to compute limitations. MASH is competitive with DPO despite this noisier reward.
> >
> > **Response to Question 4**
> >
> > We chose our training datasets (Natural Questions, HotPotQA and 2WikiMultiHopQA) because they require different search behavior and, more importantly for our research question focused on abstention, because they have different distributions for question answerability.
> >
> > To illustrate this point, we divide the test questions into 10 different buckets based on the average accuracy of parametric answers using Qwen2.5-3B-Base, i.e. without search. In the table below, we report the percentage of questions in each bucket for all datasets. For instance, the model achieves an average accuracy of 0 for 44.72% of questions and an average accuracy of 0.3 for 5.06% of questions for NaturalQuestions.
> >
> > |         Dataset         | 0     | 0.1  | 0.2  | 0.3  | 0.4  | 0.5  | 0.6  | 0.7  | 0.8  | 0.9  | 1     |
> > |------------------|-------|------|------|------|------|------|------|------|------|------|-------|
> > | Natural Questions| 44.72 | 9.62 | 6.42 | 5.06 | 4.04 | 3.07 | 3.65 | 4.04 | 3.73 | 4.40  | 11.26 |
> > | HotPotQA         | 50.11 | 9.41 | 5.91 | 4.75 | 3.86 | 3.94 | 3.54 | 3.47 | 3.71 | 3.37 | 7.92  |
> > | 2WikiMultiHopQA  | 77.26 | 8.49 | 3.77 | 2.46 | 1.97 | 1.33 | 1.22 | 0.96 | 0.85 | 0.83 | 0.85  |
> >
> > We see that all datasets have different answerability distributions. 2WikiMultiHopQA has a significantly greater number of unsolvable questions than Natural Questions and HotPotQA. The latter two have, arguably, closer unsolvable questions (Acc=0 for 44.72 vs 50.11% of questions) but differ in the number of searches needed (1-hop vs 2-hop). Therefore, we would argue that the current set of experiments already show that our method works across enough diversity of datasets.

---

> ### Author Response · Authors · 2025-11-24
> **Official Response to Reviewer b1mW Part 4**
>
> **Response to Question 2**
>
> We interpret answer rate as the percentage of questions where the model answers parametrically, abstention rate as 100-(answer rate), and recall as Abs(0), the percentage of questions where the model abstained on unanswerable questions. Note that each of these numbers can be derived from Tables 2 and Table 3. Our MASH runs have high recall while maintaining high answer rates. MASH remains competitive with DPO, with different trade-offs between answer rate and recall.
>
> | Method |  | **NQ** | | |  | **HPQA** | | |  | **2Wiki** | |
> |--------|------|------|--------|----|------|------|--------|----|------|------|--------|
> | | %Abs | %Ans | Recall | - | %Abs | %Ans | Recall | - | %Abs | %Ans | Recall |
> | OTC | 100.0 | 0.0 | 100.0 | - | 80.53 | 19.47 | 95.34 | - |96.89 | 3.11 | 97.78 |
> | MASH w/ OTC | 99.81 | 0.19 | 99.86 | - | 76.51 | 23.49 | 94.77 | - | 87.02 | 12.98 | 92.51 |
> | MASH w/ OTC-ST | 63.63 | 36.37 | 85.45 | - | 71.07 | 28.93 | 91.22 | - | 85.72 | 14.28 | 91.73 |
> | MASH w/ EXP | 64.79 | 35.21 | 85.65 | - | 76.28 | 23.72 | 94.47 | - | 88.21 | 11.79 | 93.29 |
> | 5-shot Prompting | 45.11 | 54.89 | 60.16 | - | 53.51 | 46.49 | 60.54 | - | 67.49 | 32.51 | 69.52 |
> | AFH (Absolute) | 49.86 | 50.14 | 67.71 | - | 39.51 | 60.49 | 50.42 | - | 74.41 | 25.59 | 79.5 |
> | AFH (Multisample) | 73.37 | 26.63 | 87.9 | - | 76.02 | 23.98 | 89.2 | - | 91.16 | 8.84 | 94.21 |
> | DPO | 60.43 | 39.57 | 84.48 | - | 62.55 | 37.45 | 85.91 | - | 89.66 | 10.34 | 94.11 |
>
> **We believe that our follow-up theoretical analysis and explanations address the concerns raised by the reviewer. If the reviewer finds these satisfactory, we respectfully ask that they consider raising their score to reflect these clarifications and improvements. We are happy to answer any more questions if some questions remain.**

---

> > ### Author Response · Authors · 2025-11-24
> > **References**
> >
> > **References**
> >
> > - [1] Hanning Zhang, Shizhe Diao, Yong Lin, Yi Fung, Qing Lian, Xingyao Wang, Yangyi Chen, Heng Ji, and Tong Zhang. 2024. R-Tuning: Instructing Large Language Models to Say ‘I Don’t Know’. In Proceedings of the 2024 Conference of the North American Chapter of the Association for Computational Linguistics
> >
> > - [2] Yuqing Yang, Ethan Chern, Xipeng Qiu, Graham Neubig, and Pengfei Liu. Alignment for honesty. In The Thirty-eighth Annual Conference on Neural Information Processing Systems, 2024
> >
> > - [3] Qinyuan Cheng, Tianxiang Sun, Xiangyang Liu, Wenwei Zhang, Zhangyue Yin, Shimin Li, Linyang Li, Zhengfu He, Kai Chen, and Xipeng Qiu. Can AI assistants know what they don’t know? In Forty-first International Conference on Machine Learning, 2024.
> >
> > - [4] Shangbin Feng, Weijia Shi, Yike Wang, Wenxuan Ding, Vidhisha Balachandran, and Yulia Tsvetkov. 2024. Don’t Hallucinate, Abstain: Identifying LLM Knowledge Gaps via Multi-LLM Collaboration. In Proceedings of the 62nd Annual Meeting of the Association for Computational Linguistics.
> >
> > - [5] Tianyang Xu, Shujin Wu, Shizhe Diao, Xiaoze Liu, Xingyao Wang, Yangyi Chen, and Jing Gao. 2024. SaySelf: Teaching LLMs to Express Confidence with Self-Reflective Rationales. In Proceedings of the 2024 Conference on Empirical Methods in Natural Language Processing
> >
> > - [6] Elias Stengel-Eskin, Peter Hase, and Mohit Bansal. LACIE: Listener-aware finetuning for calibration in large language models. In The Thirty-eighth Annual Conference on Neural Information Processing Systems, 2024

---

### Official Review · Reviewer_2Hdt · 2025-11-01

**Soundness:** 3
**Presentation:** 2
**Contribution:** 3
**Rating:** 6
**Confidence:** 3

**Summary:**

This work studies model's ability to abstain, including cases where models seek external search tools to answer a question. The authors propose a method named MASH seeking to minimize help seeking from external sources while abstaining when search is not available. The method uses GRPO with an additional penalty term for search. The authors train a Qwen 2.5- 3B model using MASH on 3 separate QA datasets. The authors also evaluate generalization of the best method across datasets.

**Strengths:**

The authors tackle the important practical question of abstention—the authors' definition of abstention as cases where LLMs seek external help due to the questions' answers lying outside the model's knowledge boundary is a refreshing, new perspective.

The authors propose MASH a novel method that outperforms existing baselines with a clear objective building on GRPO. The authors evaluate abstention across several scenarios: where external help is avaiable, not available, as well as generalization across datasets. Several of the insights such as the severe penality needed for models to use parameteric knoweldge (line 312), importance of SFT warmp, and difference in performance for multihop questions are all valuable insights. I also found the study of out-of-distribution generalization quite important as generalization across datasets is key to advancing abstention.

**Weaknesses:**

- clarity of the synthetic data generation pipeline: I did not find the presentation of the synthetic data generation pipeline to be very clear. A diagram or explicit example would help clarify this setp.
- Given, Qwen Base 2.5 3B is used as the basis for the specialized abstention training, it's unclear how this training affects the general capabilities of the LLM and whether abstention can be learned on top of the more commonly used chat variants. Can the authors comment on this choice and consider adding more general LLM benchmarks to capture how specialized abstention training affects general capabilities?
- How does the Qwen 2.5 3B chat model perform out of the box? This would be a key baseline to include in all the tables.

**Questions:**

- Why Exponential Decay for natural questions is different from other datasets (line 212)
- How is the Abstention Classification (Table 3) performed? Is this based on the use of search for a given question?  (line 384)
- Can you provide details on how the exact match reward used (183) is performed? Is this too strict of a criteria?

---

> ### Author Response · Authors · 2025-11-25
> **Official Response to Reviewer 2Hdt Part 1**
>
> **Response to Weakness 1**
>
> We apologize for the confusion about the synthetic data generation pipeline. The algorithm box likely complicated things and we will improve on the presentation in the next version of our paper. Our main goal is to ensure that the warm-start data has a diverse number of searches. We ensure this by constraining each generation to have a target number of searches using a very straightforward constrained decoding scheme.
>
> As an example, let’s assume we want to generate a trajectory with 2 searches.
>
> A correctly formatted trajectory would have the following format: prompt \<think\> … \</think\> \<search\> … \</search\> \<document\> … \</document\> \<think\> … \</think\> \<search\> … \</search\> \<document\> … \</document\>  \<think\> … \</think\> \<answer\> … \</answer\>.
>
> We perform the following steps to construct such a trajectory:
> - Step 1: **We append a “\<think\>” tag** to the input prompt and begin generation. This is now a standard way of running inference with recent thinking models (e.g. DeepSeek-R1’s chat template automatically adds this tag). The model generates until “\</think\>” to indicate the end of thinking.
> - Step 2: **We then automatically append a “\<search\>” tag** and continue generating until producing a “\</search\>” tag. After this, we execute the query the model specifies between \<search\> and \</search\> tags and append the retrieved documents between \<document\> and \</document\> tags.
> - Step 3: **We append a “\<think\>” tag** and allow the model to conclude its thinking, i.e. generate until it produces the “\</think\>” tag.
> - Step 4: We repeat Steps 2 and 3 to execute the second search and final thinking step.
> - Step 5: **We finally append an “\<answer\>” tag.** We stop generation when the model produces an “\</answer\>” tag.
>
> This yields a trajectory with 2 searches. Therefore, in essence, we force the model to produce a trajectory with the desired sequence of tags by appending the relevant tags at their proper places. The addition of “\<think\>”, “\<search\>” or “\<answer\>” tags to the beginning of actions additionally is a trivial modification of the regular inference procedure, which already is multi-turn.
>
> We will add this example to the paper and change our presentation of the algorithm.

---

> > ### Author Response · Authors · 2025-11-25
> > **Official Response to Reviewer 2Hdt Part 2**
> >
> > **Response to Weakness 2**
> >
> > This is a good point. We include results below that show that our specialized training does not hurt the general capabilities of the base model we use as the starting point.
> >
> > We conduct evaluations on the IFEval [1] dataset for verifiable instruction following and the MATH-Hard dataset, composed of questions of level 5 difficulty in the MATH dataset [2]. We compare our MASH w/ OTC-Strict  model’s performance against Qwen2.5-3B-Base. We use a 4-shot prompt for the MATH-Hard dataset, following convention and report results on models trained on HotPotQA in the table below.
> >
> > We do not observe any degradation in performance following our RL training. In fact, we find that our RL-trained model improves out-of-distribution, with MASH w/ OTC-Strict achieving a 1.60% improvement on IFEval and an 7.68% improvement on MATH-Hard. Note that the Base model performance we reported is similar to that reported in HuggingFace’s own Open LLM Leaderboard for these same tasks.
> >
> > | Method | IFEval | MATH-Hard |
> > |--------|--------|-----------|
> > | Qwen2.5-3B-Base | 23.60 | 15.12 |
> > | \methodname w/ OTC-ST | **25.20** | **22.80** |
> >
> > Additionally, we emphasize that any abstention training is likely to be a part of post-training pipelines, not an isolated alignment step done after the fact. The Llama-3 report [3], for instance, explicitly details their abstention pipeline for improving factuality in Section 4.3.6 as part of their post-training section, and training for tool-use is increasingly becoming an integral component of post-training today (as seen in Kimi-K2 [4]). Our experiment setup is similar to papers that perform specialized math-domain training, to isolate the effect of their approach on math capabilities of the model, even if any post-training pipeline would include other domains as well.
> >
> > *“It’s unclear [...] whether abstention can be learned on top of the more commonly used chat variants. Can the authors comment on this choice”*
> >
> > We deliberately chose to use the base model in our experiments to avoid confounders. Particularly, in our early experiments with the Qwen2.5-3B-Instruct model, we found that the model would regularly abstain even without being instructed to do so. This indicates that more recent instruction-tuned models have already undergone some safety and abstention training (although we do not know what methods or what data were used for that purpose). We take note of this in Lines 207-209. Our goal is to study the abstention capability that emerges as a result of MASH’s RL training. Using a chat model that already was trained for abstention would be a confounder.

---

> ### Author Response · Authors · 2025-11-25
> **Official Response to Reviewer 2Hdt Part 3**
>
> **Response to Weakness 3**
>
> Thanks for this suggestion. We include these results below. We conduct experiments with Qwen2.5-3B-Instruct under both the abstention setting without search tools and the search tool enabled setting.
>
> **Abstention Setting:** We evaluate the chat model for abstention under two settings:
> - Direct prompting for abstention: We include an instruction in the prompt for the model to abstain when their knowledge is lacking, using the same prompt as for the baseline abstention models in Table 3 of the paper. This, we believe, is the most natural way of eliciting abstentions from the chat model.
> - Using search as a proxy for abstention: For an apples-to-apples comparison, we simulate the setting of MASH models where we use search invocation as a proxy for abstention. We prompt the instruct models with the same prompt as the MASH models, i.e. to invoke search when its knowledge is lacking. Similar to MASH, we treat any search invocation as an abstention.
>
> We perform inference both with zero-shot prompts only containing instructions and five-shot prompts that also include exemplars. The exemplars for 5-shot are constructed to contain responses to a balanced number of unanswerable and perfectly answerable questions. More details are provided in Section B.4 of the Appendix.
>
> Our results are shown in the table below. We find that MASH w/ OTC-Strict significantly outperforms the zero- and five-shot prompted instruct model, regardless of whether abstention is instructed explicitly or induced through search. On Natural Questions, MASH w/ OTC-Strict achieves a higher accuracy than the instruct model while maintaining a similar precision value. For HotPotQA and 2WikiMultiHopQA, on the other hand, the instruct model abstains for a majority of questions, resulting in a very low answer accuracy. MASH w/ OTC-Strict achieves substantially higher accuracies while maintaining precision in these instances.
>
> | Method | NQ | | | HPQA | | | 2Wiki | |
> |--------|-----|------|--------|-----|------|----|-----|------|
> | | Accuracy | Precision | - | Accuracy | Precision | - | Accuracy | Precision |
> | **With Search Proxy**| | | | | | | | |
> | OTC | 0.00 | 0.00 | - | 12.56 | 64.51 | - | 0.75 | 24.14 |
> | MASH w/ OTC-ST | 20.89 | 57.43 | - | 17.33 | 59.89 | - | 4.64 | 32.46 |
> | 0-shot Qwen2.5-3B-Instruct (Search Prompting) | 1.51 | 35.21 | - | 0.55 | 29.29 | - | 0.09 | 5.72 |
> | 5-shot Qwen2.5-3B-Instruct (Search Prompting) | 2.95 | 37.92 | - |  2.09 | 30.60 | - | 0.13 | 6.12 |
> | **Direct Abstention**| | | | | | | | |
> | 0-shot Qwen2.5-3B-Instruct (Abstention Prompting) | 15.66 | 59.34 | - | 2.90 | 68.04 | - | 0.16 | 34.92  |
> | 5-shot Qwen2.5-3B-Instruct (Abstention Prompting) | 15.66 | 58.94 | - | 3.68 | 66.77  | - | 0.27 | 27.76 |
>
> **Inference with Search Setting:** We use the same instructions as MASH models, which describe how and when the model should search. The prompt is included in Figure 4 of the Appendix.
>
> For the search enabled setting, we find that both the zero-shot and five-shot variants severely underperform both our OTC and MASH w/ OTC-Strict checkpoints, with a ~30% performance difference in accuracy on 2WikiMultiHopQA. This shows that dedicated RL training improves search tool use capability over what the instruct model can do off-the-shelf. To our surprise, we found that the five-shot prompt underperformed the zero-shot prompt. In follow-up error analysis, we found this was due to the five-shot prompted model frequently hallucinating newer documents following search without ever reaching an answer or often failing to produce a correctly formatted answer between \<answer\> and \</answer\> tags.
>
> |Method |  | **NQ** | | |  | **HPQA** | | |  | **2Wiki** | |
> |--------|------|-----|-----|----|------|-----|-----|----|------|-----|-----|
> | | Acc↑ | TC↓ | TP↑  | - | Acc↑ | TC↓ | TP↑ | - | Acc↑ | TC↓ | TP↑ |
> | OTC | **58.95** | 1.0 | 29.47 | - | 44.76 | 0.81 | 28.64 | - | 39.59 | 1.57 | 15.32 |
> | MASH w/ OTC-ST | 56.40 | 0.64 | **38.64** | - | **53.34** | 1.10 | **32.55** | - | **46.23** | 1.64 | **19.08** |
> | 0-shot Qwen2.5-3B-Instruct | 45.63 | 1.02 | 23.26 | - | 31.44 | 1.09 | 15.55 | - | 11.02 | 1.14 | 5.21 |
> | 5-shot Qwen2.5-3B-Instruct | 35.61 | 0.98 | 19.08 | - | 30.00 | 1.08 | 15.45 | - | 14.70 | 1.31 | 6.43 |

---

> ### Author Response · Authors · 2025-11-25
> **Official Response to Reviewer 2Hdt Part 4**
>
> **Response to Question 1**
>
> We treated the $\lambda$ value in Exponential Decay as a hyperparameter and set the value following initial hyperparameter tuning experiments on the validation set.
>
> **Response to Question 2**
>
> We have two metrics under the Abstention Classification setting: Abs(0) and Delta (Abs(0) - Abs(1)). We define both of these in Section 3.2, following line 284. We will provide an explanation to expand on how these are computed, however.
>
> Before performing any training whatsoever, we first determine the knowledge boundaries of the base model. We sample 10 responses with the base model for each question using a few-shot prompt and compute average accuracies per question. This gives us information about whether a question is completely unanswerable (with an average accuracy of 0) or whether a question is always answerable (with an average accuracy of 1). This is described in lines 264-268.
>
> Abs(0) and Abs(1) then represent the abstention rate of models on questions that are completely unanswerable and always answerable respectively. Delta represents the margin between Abs(0) - Abs(1). We only evaluate these extremes because different methods we compare assume different thresholds for answerability, but there exists no consensus in prior work on what this should be. Therefore, evaluating the extremes ensures we do not favor any particular method. If you are interested in Abs(i) values for intermediate values of $i \in \\{0.1, 0.2, …, 0.9\\}$, you may refer to Part 4 of our response to Reviewer xSsG (https://openreview.net/forum?id=Dm6lP9YEsM&noteId=LaBNn59QZi) or to Section B.2 of our appendix. We find that, as we would want, models’ abstention rate decreases as the base model’s pre-computed average accuracy increases.
>
> While reporting results, we then determine whether a model abstained on a given question either by using an LLM judge for our abstention baselines or by the presence of a search call for our RL trained models.
>
> **Response to Question 3**
>
> We follow the procedure used in Search-R1’s repository (and followed by OTC as well) to compute exact match. Specifically, we remove punctuation, articles (i.e., a/an/the) and white space (beyond regular spaces) for both the model and ground-truth answer. Exact match then accepts an answer if the normalized model answer and ground-truth answer are identical.
>
> We agree that this metric can be strict and incorrectly exclude correct answers. We have two reasons for why we chose to use an exact match. Firstly, we wanted to be consistent with prior work such as Search-R1 [5] and OTC [6], both of which use exact match as their reward function. We did initially consider using the F1 score as an alternative, but decided against it as it was also unreliable. Second, while a strong LLM judge can provide a more reliable reward signal, it is infeasible to use it as an alternative to exact match during RL training given our compute limitations. As exact match is an element of the experimental design, the same training could very well be done with an LLM judge given enough compute.
>
> We do want to emphasize that exact match is only used during RL training. For all evaluation as well as in the dataset construction of abstention models, we use an LLM judge (DeepSeek-V3.1). As a result, our evaluation does not suffer from the harshness of exact match.

---

> ### Author Response · Authors · 2025-11-25
> **Official Response to Reviewer 2Hdt End**
>
> **We believe that our follow-up experiment results and explanations address the concerns raised by the reviewer. If the reviewer finds these satisfactory, we respectfully ask that they consider raising their score to reflect these clarifications and improvements. We are happy to answer any more questions if some questions remain.**
>
> **References:**
> - [1] Jeffrey Zhou, Tianjian Lu, Swaroop Mishra, Siddhartha Brahma, Sujoy Basu, Yi Luan, Denny Zhou, and Le Hou. Instruction-following evaluation for large language models. CoRR, abs/2311.07911, 2023.
> - [2] Dan Hendrycks, Collin Burns, Saurav Kadavath, Akul Arora, Steven Basart, Eric Tang, Dawn Song, and Jacob Steinhardt. Measuring mathematical problem solving with the math dataset. NeurIPS, 2021.
> - [3] Abhimanyu Dubey, Abhinav Jauhri, Abhinav Pandey, Abhishek Kadian, Ahmad Al-Dahle, Aiesha Letman, Akhil Mathur, Alan Schelten, Amy Yang, Angela Fan, et al. 2024. The llama 3 herd of models. arXiv preprint arXiv:2407.21783.
> - [4] Kimi Team, et al. 2025. "Kimi-K2: Open agentic intelligence." arXiv preprint arXiv:2507.20534.
> - [5] Bowen Jin, Hansi Zeng, Zhenrui Yue, Jinsung Yoon, Sercan O Arik, Dong Wang, Hamed Zamani, and Jiawei Han. Search-R1: Training LLMs to reason and leverage search engines with reinforcement learning. In Second Conference on Language Modeling, 2025.
> - [6] Hongru Wang, Cheng Qian, Wanjun Zhong, Xiusi Chen, Jiahao Qiu, Shijue Huang, Bowen Jin, Mengdi Wang, Kam-Fai Wong, and Heng Ji. Acting less is reasoning more! Teaching model to act efficiently. arXiv preprint arXiv:2504.14870, 2025a.

---

### Official Review · Reviewer_wZVe · 2025-11-02

**Soundness:** 3
**Presentation:** 3
**Contribution:** 3
**Rating:** 6
**Confidence:** 3

**Summary:**

This paper introduces MASH (Modeling Abstention via Selective Help-seeking), which is a reinforcement learning framework that trains LLMs to selectively invoke external search tools under a pay-per-search penalty. The key idea is that selective help-seeking implicitly learns abstention behavior: if the model wants to search, that indicates it cannot answer with parametric knowledge. Removing the search tool at inference thus converts the model into an abstention model.

**Strengths:**

* Clever reframing of help-seeking as a proxy for abstention. Avoids need for labeled “known/unknown” training data.
* The writing is clear. The paper is easy to follow.

**Weaknesses:**

* The link between search invocation and calibrated abstention is intuitive but not formally analyzed.
* Focuses solely on short-form QA; unclear applicability to reasoning-heavy or generative tasks, e.g., mathematical problems.

**Questions:**

None

---

> ### Author Response · Authors · 2025-11-25
> **Official Response to Reviewer wZVe Part 1**
>
> We thank the reviewer for their comments. We will address the main points below.
>
> **Response to Weakness 1**
>
> We thank the reviewer for their suggestion to place a formal analysis of our proxy objective. Its inclusion will improve the soundness of our paper.
>
> Let $\pi_{\theta^{\star}}(\tau \mid q, H)$
> be the probability of trajectory $\tau$ given question $q$ and helper $H$, and let $N_s(\tau)$ be the number of searches in $\tau$. Then the optimal policy will answer question $q$ parametrically if and only if $E_{\\tau \\sim \\pi_{\\theta^{\star}}}\\!\\left[r_{\\mathrm{acc}}(y,\\tau)*r_{\\mathrm{help}}(q,\\tau)\\mid N_s(\\tau)=0\\right] \\ge E_{\\tau \\sim \\pi_{\\theta^{\star}}}\\!\\left[r_{\\mathrm{acc}}(y,\\tau)*r_{\\mathrm{help}}(q,\\tau)\\mid N_s(\\tau)=i\\right]$  for all $i>0$.
>
>
>
>
> **We provide a formal proof of this in Section A of the Appendix.** In plain English, however, the optimal policy will answer question $q$ parametrically if and only if its average reward when answering parametrically is greater than or equal to the expected reward it would obtain when answering with external help. To concretize this with an example, consider the oracle helper setting and let the help penalty be $r_{help}(y, \tau) = \lambda^{N_s(\tau)}$, where $0 < \lambda < 1$ is a hyperparameter controlling the severity of the penalty and $N_s(\tau)$ is the number of searches performed by $\tau$. Then, the model will answer parametrically if its average accuracy is greater than or equal to $\lambda$. Particularly, it will exclusively answer parametrically if its average accuracy is greater than $\lambda$.
>
> For regular helpers, this boundary will depend both on the strength of the retrieval penalty and on the reliability of the help. Therefore, during our RL training, the model simultaneously learns how to seek external help, and when to do so based on its parametric knowledge and its implicit estimate of the helper's reliability.
>
> **Response to Weakness 2**
>
> We follow prior literature on abstention [1,2,3,4,5,6] and use short-form QA for evaluation. We assume that prior works make this choice because reliable evaluation of long-form answer correctness is still an open problem. While extending our framework on these other settings or other domains such as math or coding is interesting, and will likely invite unique challenges, we assert that this is beyond the scope of our work.

---

> ### Author Response · Authors · 2025-11-25
> **Official Response to Reviewer wZVe Part 2**
>
> **We believe that our follow-up theoretical analysis and explanations address the concerns raised by the reviewer. If the reviewer finds these satisfactory, we respectfully ask that they consider raising their score to reflect these clarifications and improvements. We are happy to answer any more questions if some questions remain.**
>
> **References**
>
> - [1] Hanning Zhang, Shizhe Diao, Yong Lin, Yi Fung, Qing Lian, Xingyao Wang, Yangyi Chen, Heng Ji, and Tong Zhang. 2024. R-Tuning: Instructing Large Language Models to Say ‘I Don’t Know’. In Proceedings of the 2024 Conference of the North American Chapter of the Association for Computational Linguistics
>
> - [2] Yuqing Yang, Ethan Chern, Xipeng Qiu, Graham Neubig, and Pengfei Liu. Alignment for honesty. In The Thirty-eighth Annual Conference on Neural Information Processing Systems, 2024
>
> - [3] Qinyuan Cheng, Tianxiang Sun, Xiangyang Liu, Wenwei Zhang, Zhangyue Yin, Shimin Li, Linyang Li, Zhengfu He, Kai Chen, and Xipeng Qiu. Can AI assistants know what they don’t know? In Forty-first International Conference on Machine Learning, 2024.
>
> - [4] Shangbin Feng, Weijia Shi, Yike Wang, Wenxuan Ding, Vidhisha Balachandran, and Yulia Tsvetkov. 2024. Don’t Hallucinate, Abstain: Identifying LLM Knowledge Gaps via Multi-LLM Collaboration. In Proceedings of the 62nd Annual Meeting of the Association for Computational Linguistics.
>
> - [5] Tianyang Xu, Shujin Wu, Shizhe Diao, Xiaoze Liu, Xingyao Wang, Yangyi Chen, and Jing Gao. 2024. SaySelf: Teaching LLMs to Express Confidence with Self-Reflective Rationales. In Proceedings of the 2024 Conference on Empirical Methods in Natural Language Processing
>
> - [6] Elias Stengel-Eskin, Peter Hase, and Mohit Bansal. LACIE: Listener-aware finetuning for calibration in large language models. In The Thirty-eighth Annual Conference on Neural Information Processing Systems, 2024

---

> > ### Comment · Reviewer_wZVe · 2025-11-26
> >
> > Thank you for your response and for all the effort you have put into the work. Most of my concerns have been resolved. After reading the other reviewers’ comments and your replies, and reflecting on them, I have decided to maintain my overall positive recommendation. However, since I am not an expert in large language models, I have lowered my confidence in this assessment.

---

### Official Review · Reviewer_xSsG · 2025-11-07

**Soundness:** 2
**Presentation:** 2
**Contribution:** 3
**Rating:** 4
**Confidence:** 5

**Summary:**

This paper explores an interesting insight: that training LLMs for selective help-seeking (knowing when to search) naturally induces abstention behavior (knowing when to say "I don't know"). The authors propose MASH, which uses reinforcement learning with pay-per-search penalties to train models that invoke search only when needed. The key observation is that when search access is removed at inference, search invocations become abstention signals. The authors demonstrate improvements over baselines on three QA datasets and analyze the behavior of the trained LLMs.

**Strengths:**

* **Novel perspective**. The paper reframes the abstention problem from direct boundary detection through tool use. This shift from "teaching what not to know" to "learning when to seek help" opens a different approach to modeling uncertainty in LLMs. Unlike existing abstention methods that require oracle knowledge of model capabilities to construct training data, This method discovers knowledge boundaries through RL optimization. The model self-identifies its limitations via the help-seeking reward signal, making it more scalable and realistic.
* **Empirical validation of non-obvious transfer**. The successful transfer from search behavior to abstention is insightful. The knowledge boundary identification ability without privileged information required can achieve a comparable performance with doing SFT on a specially curated abstention dataset.

**Weaknesses:**

1. **Necessity of warm-start trajectory construction**. The paper lacks justification for the complex warm-start procedure over simpler alternatives, such as rejection sampling with format constraints. Table 4 shows models trained without warm-start, but no direct comparison between warm-start trajectory construction and rejection sampling is provided. The rationale for using a base model rather than an instruct model for synthetic data generation remains unclear and may introduce unnecessary complexity.
2. **Inadequate handling of partial knowledge and reward exploitation**. MASH primarily addresses extreme cases (Abs(0) and Abs(1)) while neglecting the critical middle ground where models have partial knowledge. Several issues arise: (1) In multi-hop reasoning, models may just happen to generate intermediate entities without retrieval (Pass@K). It causes MASH to penalize all other trajectories in the GRPO group, leading to unstable RL training. (2) For multiple-choice or binary questions, models can achieve rewards through random guessing with incorrect reasoning, a problem observed in Search-R1 that MASH's strict penalties may exacerbate. (3) The aggressive penalty structure makes integration with other reward signals challenging without careful balancing.
3. **Limited benchmark coverage**. The evaluation omits MuSiQue, a widely used benchmark that supports at most 4-hop reasoning, which would better test the approach's scalability for complex multi-hop queries.
4. **Insufficient model scale evaluation**. Experiments exclusively use Qwen2.5-3B-base, leaving open the question of whether findings generalize to larger models (7B, 14B) or newer architectures (Qwen3), where different dynamics may emerge.
5. **Incomplete baseline analysis**. Table 2 omits Search-R1's detailed search distribution. The consistent TC=3.0 for Search-R1 on multi-hop tasks (Table 1) seems weird. The model should sometimes answer directly or perform additional searches if it fails to generate appropriate queries.
6. **Presentation clarity issues**. The paper's flow and readability need improvement. For instance, the abstract's key idea is difficult to follow. Moreover, the paper suffers from excessive use of dashes, disrupting readability.

**Questions:**

Line 177. What is "random correct?" Does it mean rolling out multiple trajectories that contain the correct answer, and then randomly picking one? And why use "shortest answer" if we cannot get one correct trajectory?

---

> ### Author Response · Authors · 2025-11-24
> **Official Response to Reviewer xSsG Part 1**
>
> Thank you for your comments. We address your main points below, in order of priority. (We apologize for the length of the response but it was unavoidable to address each point that the reviewer raised).
>
> **Insufficient model scale evaluation:**
>
> In our paper, we restrict our analysis to the Qwen-2.5-3B model in order to be able to cover 3 different QA datasets, 3 different reward formulations, and include 6 baseline methods for comparisons as well as additional ablations (without warm start, out-of-distribution generalization analysis). These results provide convincing evidence for our method’s performance.
>
> As suggested by the reviewer, we conduct additional experiments to show that our findings transfer to other scales (we use Qwen2.5-7B-Base) and model families (Qwen3-4B-Base). Note that RL training for both our approach and baseline OTC takes ~3-4 days to train on one H100, so we restrict our analysis to HotPotQA for the rebuttal (and train all models for 300 steps as opposed to 400 in the original paper).
>
> | **Method** | **Qwen2.5-7B-Base** | | | | **Qwen3-4B-Base** | | |
> |------------|---------------------|----------|----------|----|---------------------|----------|----------|
> | | Acc | TC | TP | - | Acc | TC | TP |
> | OTC  | 51.52 | 1.00 | 25.76 |  - | 49.11 | 1.00 | 24.55 |
> | MASH w/ OTC-ST | **55.13** | 1.18 | **35.37** |  - | **51.45** | 0.90 | **34.13** |
>
> MASH reports similar gains over OTC with these new models as with Qwen-2.5-3B. MASH w/ OTC-Strict outperforms the OTC in terms of both Accuracy and Tool Productivity, reporting ~10% improvement in the latter. Crucially, MASH w/ OTC-Strict outperforms the OTC baseline for Qwen3-4B on Accuracy despite searching less.
>
> | **Method** | **Qwen2.5-7B-Base** ||||| **Qwen3-4B-Base** ||||
> |------------|------|------|------------|---------|----|------|------|------------|---------|
> |            | Acc  | Prec | Abs(0)↑    | Delta↑  | - | Acc  | Prec | Abs(0)↑    | Delta↑  |
> | OTC | 0.00 | -- | 100.00 | 0.00 | - | 0.00 | -- | 100.00 | 0.00 |
> | MASH w/ OTC-ST | 20.98 | 60.67 | 87.14 | 64.51 | - | 20.96 | 53.59 | 81.63 | 67.07 |
> | 5-shot Prompting | 23.06 | 34.17 | 39.45 | 24.18 | - | 17.58 | 34.37 | 59.76 | 43.72 |
> | AFH (Absolute) | 25.52 | 36.73 | 40.3 | 27.93 | - | 13.44 | 47.97 | 82.81 | 41.5 |
> | AFH (Multisample) | 17.68 | 51.36 | 82.00 | 53.22 | - | 7.25 | 66.54 | 96.54 | 36.51 |
> | DPO | 24.35 | 48.25 | 72.8 | 60.4 | - | 16.72 | 60.84 | 92.38 | 75.87 |
>
> We also see similarly strong results on abstention. Qwen2.5-7B and Qwen3-4B both strongly outperform the 5-shot Prompting and SFT-based baselines. Qwen2.5-7B achieves competitive accuracies while maintaining up to 26% higher precision values, while Qwen3-4B outperforms all other baselines in terms of accuracy. Finally, similar to the Qwen2.5-3B-Base results, MASH w/ OTC-Strict is competitive with DPO, with different accuracy-precision trade-offs (recall that precision can be “gamed” by over-abstaining at the cost of accuracy). For Qwen3-4B, MASH outperforms DPO on accuracy at the cost of slightly lower precision, while the opposite is true for Qwen2.5-7B.

---

> > ### Author Response · Authors · 2025-11-24
> > **Official Response to Reviewer xSsG Part 2**
> >
> > **Limited benchmark coverage.** Thanks for bringing this up. We did initially consider MuSiQue for our experiments but found that dataset to be unsuitable for studying abstention. We explain this in detail below.
> >
> > First, we agree that MuSiQue is a challenging dataset with multi-hop reasoning phenomena that would be interesting to study. However, our goal is to study whether models can make abstention decisions aligned with their parametric knowledge. For MuSiQue, the Qwen-2.5-3B reports only ~4% accuracy when answering parametrically. This means that any abstention model trained on MuSiQue should abstain for almost the entire dataset, making this dataset unsuitable for studying abstention.
> >
> > While we cannot run abstention analysis for MuSiQue, we assert that MASH does outperform the OTC baseline in the inference setting with search tool access. We replicate Table 1 and show search distributions for MuSiQue below. MASH achieves ~9% higher accuracy than the OTC baseline and produces a diverse number of searches, going up to 4 while the OTC baseline converges to 1 search.
> >
> > **Replication of Table 1:**
> > | **Method** | **MuSiQue** |||
> > |------------|-------------|-----|------|
> > |            | Acc↑        | TC↓ | TP↑  |
> > | OTC | 14.22 | 1.00 | 7.12 |
> > | MASH w/ OTC-ST | **23.67** | 2.23 | **8.08** |
> >
> > **Search Distribution for OTC and MASH w/ OTC-ST on MuSiQue**
> > | **Method** | **Search Distribution** |||||
> > |------------|------|------|------|------|------|
> > |            | 0    | 1    | 2    | 3    | 4+    |
> > | OTC | 0.3 | 99.6 | 0.0 | 0.0 | 0.0 |
> > | MASH w/ OTC-ST | 8.1 | 5.0 | 52.0 | 27.2 | 7.8 |

---

> > > ### Author Response · Authors · 2025-11-24
> > > **Official Response to Reviewer xSsG Part 3**
> > >
> > > **Necessity of warm-start trajectory construction.**
> > >
> > > We believe there has been some confusion about understanding our warm-start process and its complexity. To clarify, our warm-start algorithm essentially implements a very simple constrained decoding scheme to ensure that trajectories contain a target number of searches. We will improve the presentation of the algorithm to fix this issue.
> > >
> > > We want to emphasize that the main goal of warm-start is **not** to teach models the right format. In fact, our models trained without warm start (i.e. baseline OTC and all models in Table 4 of the paper) learn the correct format very early in the training. This is what the rejection sampling process suggested by the reviewer would address. Instead, the aim is to teach models to generate trajectories with a diverse number of searches. The reason why OTC underperforms MASH is because it converges to one particular search behavior (e.g. 1 search for natural questions) instead of exploring diverse search behaviors during training.
> > >
> > > Our pipeline additionally helps eliminate any confounders. The warm-start data is constructed so that there is no link between number of searches and answerability; recall that we randomly sample the number of searches per input. This helps us ensure that the model learns any abstention capabilities only via RL. Our use of a base model for warm-start data generation is motivated by the same concern. In fact, our initial explorations show that instruct models, likely because of their safety training, can already abstain even without explicit prompting. Using a base model avoids these risks altogether.

---

> > > > ### Author Response · Authors · 2025-11-24
> > > > **Official Response to Reviewer xSsG Part 4/1**
> > > >
> > > > **Inadequate handling of partial knowledge and reward exploitation.**
> > > >
> > > > *“MASH primarily addresses extreme cases (Abs(0) and Abs(1)) while neglecting the critical middle ground where models have partial knowledge.”*
> > > >
> > > > **Response:** We want to clarify that Abs(0) and Abs(1) values are a component of our evaluation, not of the MASH method itself. MASH’s training procedure makes no assumptions about whether a question is answerable a priori. As we explain in the paper, we choose to focus on Abs(0) and Abs(1) as they evaluate cases where the abstention decision is straightforward and does not unfairly favor any baseline or reward formulation. However, models do demonstrate highly interpretable trends for intermediate values as well. We show below Abs(i) values for each  $i \in {0, 0.1, 0.2, …, 0.9, 1.0}$ for MASH w/ OTC-Strict in the table below.
> > > >
> > > > | Method | Abs(0) | Abs(0.1) | Abs(0.2) | Abs(0.3) | Abs(0.4) | Abs(0.5) | Abs(0.6) | Abs(0.7) | Abs(0.8) | Abs(0.9) | Abs(1) |
> > > > |--------|-------|-------|-------|-------|-------|-------|-------|-------|-------|-------|-------|
> > > > | Natural Questions | 85.45 | 75.14 | 67.27 | 52.85 | 53.62 | 51.13 | 46.40 | 33.50 | 27.78 | 27.37 | 19.29 |
> > > > | HotPotQA | 91.22 | 82.26 | 67.44 | 56.96 | 46.67 | 40.67 | 38.26 | 40.66 | 36.55 | 28.30 | 30.92 |
> > > >
> > > > As we would want, the model abstention rate decreases as the base model’s average accuracy increases. On Natural Questions, for instance, MASH w/ OTC-Strict has an Abs(0) of 85.43%, meaning that it abstains 85.43% of the time for questions where the base model was always incorrect. The abstention rate decreases to 51.13% for questions with an average accuracy of 0.5 (Abs(0.5)) and achieves its lowest value for Abs(1). This shows that the models’ search/abstention behavior is affected in a highly interpretable manner “in the middle ground where models have partial knowledge.” Tables featuring all models can be found in Appendix B.2.
> > > >
> > > > Furthermore, we emphasize that our reward structure does handle cases where the model has partial knowledge in a multi-hop setting. Assume that the model knows only one fact in a two-hop question. It will maximize its reward only if it performs one search for the unknown fact, as the penalty is dependent on the number of searches, and answers parametrically for the other. As such, an optimal policy would also recognize partial knowledge.
> > > >
> > > > *“In multi-hop reasoning, models may just happen to generate intermediate entities without retrieval (Pass@K). It causes MASH to penalize all other trajectories in the GRPO group, leading to unstable RL training.”*
> > > >
> > > > **Response:** If we understand correctly, the reviewer is pointing out a scenario where a model’s knowledge is uncertain but it accidentally gets a sub-question correct in one of the K samples. Concretely, assume the expected accuracy for answering the first sub-question q parametrically and correctly, i.e. E[Param-Acc(q)], is low, say 0.2. The reviewer argues that there will be a trajectory that searches for q, and answers the overall question correctly. But there may also exist a trajectory that answers q parametrically and is accidentally correct (possible as E[Param-Acc(q)] > 0). They argue that this will lead to unstable RL training.
> > > >
> > > > We disagree. First, in the above scenario itself, the model will also see other trajectories where q is answered parametrically and incorrectly, as E[Param-Acc(q)] is only 0.2 in our example. Therefore, for 0.2, it will likely see more trajectories where answering q parametrically leads to an incorrect answer than correct, and also see examples where searching for q leads to a correct answer. It will accordingly adjust its value for the search action. For a q with a different E[Param-Acc(q)], say = 0.8, the model will conversely see more reward (in expectation) when answering q parametrically than searching and incurring a penalty. Overall, the model will encounter different questions with varying parametric accuracies during training; the goal of our RL training is to learn to invoke search with different probabilities for these in order to maximize the expected reward. (Note that we only include estimates of expected accuracy of subquestions here for illustration. We do not directly reward sub-questions).
> > > >
> > > > Empirically, our RL training runs are stable and we report evidence of alignment of search/abstention behavior with parametric knowledge (Table 3 in the paper). Both this explanation and empirical evidence addresses the reviewer’s concern.

---

> > > > > ### Author Response · Authors · 2025-11-24
> > > > > **Official Response to Reviewer xSsG Part 4/2**
> > > > >
> > > > > *“For multiple-choice or binary questions, models can achieve rewards through random guessing with incorrect reasoning, a problem observed in Search-R1”*
> > > > >
> > > > > **Response:** In our experiments, we found that the Search-R1 models actually max out the number of allowable searches (detailed explanation in the next question). Therefore, we did not encounter this random guessing behavior empirically. This also follows intuitively as the reward enforces no pressure for the model to learn to answer parametrically. Additionally, we could not find this claim in the Search-R1 paper (either in the main or the appendix). Could you point us to the text so we can better understand their argument, since we both intuitively and empirically disagree with it.
> > > > >
> > > > > Moreover, the reward hacking behavior the reviewer describes is a flaw shared by all baseline methods, including OTC and DPO/AFH (the latter of which uses a 0.1 threshold when constructing gold training data). In fact, this is also a flaw of any math model RL trained with only the correctness reward in a multiple choice setting; the model would be similarly susceptible to guessing without learning good reasoning strategies. The focus of this paper is completely different; we do not try to fix all issues with outcome-based RL training.
> > > > >
> > > > >
> > > > > *“The aggressive penalty structure makes integration with other reward signals challenging without careful balancing.”*
> > > > >
> > > > > **Response:** First, we emphasize that the penalty structure does not need to be aggressive. Our paper already includes experiments with 3 different penalty formulations with varying levels of “aggressiveness” to showcase this point. Model performance for HotPotQA and 2WikiMultiHopQA was similar across all formulations, while Natural Questions reported the best performance with the severe penalty option.
> > > > >
> > > > > We also emphasize that adding an additional penalty to the correctness reward should not be treated as a flaw by itself. Several published papers in the last few months penalize other “undesirable” behaviors similarly, e.g. length or format. This modification to RL training is an established method in literature.

---

> ### Author Response · Authors · 2025-11-24
> **Official Response to Reviewer xSsG Part 5**
>
> **Incomplete baseline analysis:**
>
> Good catch! For Search-R1, we set the maximum number of allowed searches to 3. We describe this in Appendix C.2, but understand how this could be confusing. We will add this detail to the main text as well.
>
> As Search-R1’s reward formulation produces no pressure for models to answer questions directly, correct parametric and search-augmented answers both receive the same reward. Performing multiple searches leads to correct answers more reliably than parametric answers and models converge to consistently searching as a result.
>
> An additional minor point is as follows: if models generate an additional <search> tag after already executing the maximum number of searches, our pipeline will not execute that search or return new documents. Therefore, even though, technically, the average tool calls can be greater than 3, we clamp the reported number to 3 for clarity. We will explain this and include Search-R1 in Table 2 in the next version of the paper.
>
>
> **Response to Question**
>
> We roll out multiple trajectories during warm-start generation. If there are >1 correct trajectory for a question, we randomly sample one of these correct trajectories. This is what we mean by “random correct”. If all trajectories have incorrect answers, we pick the trajectory with the shortest answer to avoid unintentionally biasing the models towards long sentence-length answers as our datasets are short-form QA. We will clarify this.
>
> **We believe that our follow-up experiment results and explanations address the concerns raised by the reviewer. If the reviewer finds these satisfactory, we respectfully ask that they consider raising their score to reflect these clarifications and improvements. We are happy to answer any more questions if some questions remain.**

---

> > ### Comment · Reviewer_xSsG · 2025-11-27
> >
> > Thanks for the rebuttal, which has addressed most of my concerns.
> >
> > This work offers a novel perspective by reframing the abstention problem from direct boundary detection to tool use, which I find to be a valuable contribution. The analysis is well-executed.
> >
> > However, I remain concerned about the proposed method. While I agree with the authors' argument that the goal is not to teach models the correct format, the proposed approach does not demonstrate significant improvements or a novel solution over naive sampling baselines. I understand this paper may not focus on the methodological aspects and may also leave room for future research.
> >
> > Therefore, I am raising my score to 6. I am also adjusting my confidence to 4, as further discussion has revealed some nuances I may have initially overlooked.

---

> > > ### Author Response · Authors · 2025-12-02
> > > **Response to Reviewer xSsG**
> > >
> > > We thank the reviewer for their response. We are glad that our additional experiments addressed your comments about generalization to additional models and datasets, and that our explanations helped reveal further nuances.
> > >
> > > We want to emphasize that the main contribution of our work is a method to extract abstention decisions from LLMs without explicitly training for abstention or requiring ground truth abstention training data. This is a fundamentally new way of obtaining abstention decisions from LLMs, and does so while simultaneously improving the help-seeking capabilities of models.
> > >
> > > Therefore, we disagree with your comment that the “proposed approach does not demonstrate significant improvements over naive sampling baselines [...]”. This incorrectly reduces the entirety of our contribution to the synthetic data generation pipeline. We want to clarify that the synthetic data generation pipeline is a pre-RL step and is only a minor component of the overall contribution designed to address a particular error mode observed during RL.

---

### Author Response · Authors · 2025-12-02
**General Comment Summarizing Rebuttal Part 1**

We thank all reviewers for their comments. Each of our rebuttals were lengthy to address the reviewers’ comments with the nuance they deserved. We present a summary of our rebuttals for the benefit of the AC.

Reviewers uniformly recognized our key contribution that training models to selectively seek help (such as by calling search tools) also produced abstention models by proxy as exciting, describing it as “a refreshing, new perspective” (Reviewer 2Hdt) “elegantly link[ing] tool-use efficiency to abstention behavior” (Reviewer b1mW). The reviewers who engaged in discussion either increased their score from 4 to 6 (Reviewer xSsG) or “maintained their overall positive recommendation” (Reviewer wZVe).

As Reviewers b1mW and 2Hdt were not able to respond, we will summarize our rebuttals to them.

**Summary of Rebuttal to Reviewer b1mW (did not respond to rebuttal):**

Reviewer b1mW did not respond before discussion was frozen, **but noted that they would raise their score if their concerns were resolved.** We believe we addressed their key questions in our rebuttal.

The reviewer’s main concern was that the paper is missing a theoretical account for why our approach “should produce a stable abstention boundary” and that “the mechanism remains unclear and appears incidental”. **In response, we added a formal proof in Appendix A of our paper.**  To briefly explain, we train an LLM to use search tools to solve QA tasks with RL. Our reward has the structure $r=r_{acc}*r_{penalty}$, i.e. we reward the model for accuracy but penalize it for search use. We posit that at inference, any search invocation can be used as a proxy for abstention. Empirically, our paper showed that this does result in a performant abstention model, with abstention (i.e. search use) aligned with parametric knowledge.

We add a proof to show that the optimal policy will answer parametrically (i.e. not use search) for a particular question if and only if the expected accuracy when producing parametric answers is greater than or equal to the expected reward (with penalty) when invoking search. This expected reward is then dependent on the strength of the search penalty and of the search tool. This proves that this mechanism is not “incidental” and addresses the reviewer's concern.

The reviewer’s second concern was that in our experiments with an oracle helper, “the optimal policy is trivially to always ask for help, since help deterministically returns the gold answer.” **We show that this claim is incorrect using our proof from Appendix A.** The optimal policy should ask for help only if its average accuracy when answering parametrically is below $\lambda$, the penalty associated with seeking help. The reviewer further expressed doubt about the equivalence we made between our oracle helper setting and explicit RL training for abstention with a ternary reward. **We provide both an empirical and theoretical substantiation for our claim.** We conduct experiments with the ternary reward and show that models uniformly converge to always abstaining within 25 steps, similar to our oracle helper setting. We also show a formal equivalence between the two settings, demonstrating that both have identical ternary reward structures.

The final concern raised by the reviewer was that our experiments are restricted to QA datasets. In our response, we note that we follow prior literature on abstention [1, 2, 3, 4, 5, 6] by using short-form QA for evaluation. Extending our framework to other settings is beyond the scope of our work. Our complete response to this is here: https://openreview.net/forum?id=Dm6lP9YEsM&noteId=7g17gmdfeD

**Summary of Rebuttal to Reviewer 2Hdt (did not respond to rebuttal):**

We ran additional experiments to address two of the reviewer’s questions.
- The reviewer asked how our abstention training affected models’ general capabilities. We ran experiments on two additional tasks, instruction following and math, and showed that our training did not degrade these capabilities (see detailed response here: https://openreview.net/forum?id=Dm6lP9YEsM&noteId=l7DgmuoAWn). We also included a discussion of why our choice of running experiments on the base models helps avoid confounders compared to using instruct models.
- The reviewer also inquired about the performance of instruct models off-the-shelf. We included extensive experiments under both zero- and few-shot settings and show that **our trained models significantly outperform the Qwen2.5-3B-Instruct model, both for abstention and tool use.**

The only remaining weakness raised by the reviewer was the presentation of our synthetic data generation pipeline. We clarify the approach in the response and will improve its presentation in the main paper as well. We believe these additional experiments address all of the concerns raised by this reviewer.

---

> ### Author Response · Authors · 2025-12-02
> **General Comment Summarizing Rebuttal Part 2**
>
> **Summary of Reviewer xSsG and wZVe’s responses.**
>
> **Reviewer xSsG increased their score to a 6 following our rebuttal, noting:** *“Thanks for the rebuttal, which has addressed most of my concerns. This work offers a novel perspective by reframing the abstention problem from direct boundary detection to tool use, which I find to be a valuable contribution. The analysis is well-executed. [...] Therefore, I am raising my score to 6. I am also adjusting my confidence to 4, as further discussion has revealed some nuances I may have initially overlooked.”*
>
> The reviewer additionally adds that they *“remain concerned about the proposed method. While I agree with the authors' argument that the goal is not to teach models the correct format, the proposed approach does not demonstrate significant improvements or a novel solution over naive sampling baselines. I understand this paper may not focus on the methodological aspects and may also leave room for future research.”*
>
> The discussion period was frozen before we could engage with the reviewer further. **We emphasize that this comment mischaracterizes the main contribution of our work.** It reduces our contribution to the synthetic data generation pipeline, which is completely incorrect.
>
> Our main contribution is a method to extract abstention decisions from LLMs without explicitly training for abstention. Our method uses RL to train a model for selective help-seeking (e.g., search tool use), which is then used as a proxy for abstention at inference. This is a fundamentally new way of obtaining abstention decisions from LLMs, and does so while simultaneously improving the help-seeking capabilities of models. The synthetic data generation pipeline is a **pre-RL step** and is only a minor component of the overall contribution. It is designed to address a particular error mode observed during RL. We hope that the AC would take this clarification into account when interpreting the reviewer’s comments.
>
> **Reviewer wZVe maintained their score of 6, noting:** *“Thank you for your response and for all the effort you have put into the work. Most of my concerns have been resolved. After reading the other reviewers’ comments and your replies, and reflecting on them, I have decided to maintain my overall positive recommendation. However, since I am not an expert in large language models, I have lowered my confidence in this assessment.”*
>
> **References:**
> - [1] Hanning Zhang, Shizhe Diao, Yong Lin, Yi Fung, Qing Lian, Xingyao Wang, Yangyi Chen, Heng Ji, and Tong Zhang. 2024. R-Tuning: Instructing Large Language Models to Say ‘I Don’t Know’. In Proceedings of the 2024 Conference of the North American Chapter of the Association for Computational Linguistics
>
> - [2] Yuqing Yang, Ethan Chern, Xipeng Qiu, Graham Neubig, and Pengfei Liu. Alignment for honesty. In The Thirty-eighth Annual Conference on Neural Information Processing Systems, 2024
>
> - [3] Qinyuan Cheng, Tianxiang Sun, Xiangyang Liu, Wenwei Zhang, Zhangyue Yin, Shimin Li, Linyang Li, Zhengfu He, Kai Chen, and Xipeng Qiu. Can AI assistants know what they don’t know? In Forty-first International Conference on Machine Learning, 2024.
>
> - [4] Shangbin Feng, Weijia Shi, Yike Wang, Wenxuan Ding, Vidhisha Balachandran, and Yulia Tsvetkov. 2024. Don’t Hallucinate, Abstain: Identifying LLM Knowledge Gaps via Multi-LLM Collaboration. In Proceedings of the 62nd Annual Meeting of the Association for Computational Linguistics.
>
> - [5] Tianyang Xu, Shujin Wu, Shizhe Diao, Xiaoze Liu, Xingyao Wang, Yangyi Chen, and Jing Gao. 2024. SaySelf: Teaching LLMs to Express Confidence with Self-Reflective Rationales. In Proceedings of the 2024 Conference on Empirical Methods in Natural Language Processing
>
> - [6] Elias Stengel-Eskin, Peter Hase, and Mohit Bansal. LACIE: Listener-aware finetuning for calibration in large language models. In The Thirty-eighth Annual Conference on Neural Information Processing Systems, 2024

---

### Meta-Review · Area_Chair_YxQV · 2025-12-29

**Summary:**

The generalizability of of the proposed method over helpers with diverse performance levels still (Reviewer b1mW) remains and critical. In particular,
the authors' theoretical justification in Appendix A does not provide a clear definition on an optimal policy and depends on Assumption 2, which is not trivial as it depends on a helper. Currently, the proposed abstention method heavily relies on the performance of the helper, so at least authors need to show empirical justification whether the claim maintains over helpers with diverse performance levels, which is missing. I expect that this cannot be addresable during the rebuttal period, voting for rejection.

**Reviewer Concerns:**

**Reviewer xSsG**:
This reviewer raised six concerns and they are mostly addressed as follows:
1. Necessity of warm-start trajectory construction – explained and acknowledged by the reviewer
2. Inadequate handling of partial knowledge and reward exploitation – explained and acknowledged by the reviewer
3. Limited benchmark coverage – addressed by providing additional experiments on MuSiQue, showing similar trends, and acknowledged by the reviewer
4. Insufficient model scale evaluation  – addressed by providing additional experiments on Qwen2.5-7B-Base / Qwen3-4B-Base, showing similar trends, and acknowledged by the reviewer
5. Incomplete baseline analysis – explained and acknowledged by the reviewer
6. Presentation clarity issues – not addressed but minor.

The reviewer raised incremental performance gain (partially related to concerns 3 and 4), and I agree that this concern is still outstanding. In particular, the proposed method is incremental (i.e., added SFT pipeline and improved a reward function from the baseline (Wang et al., 2025a). The authors disagreed with this saying that ``the main contribution of our work is a method to extract abstention decisions from LLMs without explicitly training for abstention or requiring ground truth abstention training data.’’. This is true but at the point of calling tools, it is unnecessary to remove tool calling and replace it into an abstention. Instead, it is more beneficial to provide abstentions for tool-calling LLMs.

**Reviewer wZVe**:
This reviewer raised two concerns and they are addressed as follows:
1. Theoretical connection between search invocation and abstention – conducted a loosely related theoretical analysis which is not directly related to abstention.
2. Missing evaluation on advanced tasks with heavy reasoning – addressed claiming that related papers are evaluated on short-form QA

No outstanding unaddressed concerns.


**Reviewer 2Hdt**:
This reviewer raised six concerns and they are partially addressed as follows:
1. Missing details on synthetic data generation pipeline – additional details are provided.
2. Side-effect of abstention learning on the capability of a base model – empirically shown no side-effects on general performance.
3. Missing Qwen 2.5 3B chat model as a baseline – empirically demonstrated the results and following the same trend as a Qwen 2.5. 3B base model.
4. Unclearn Hyper-parameter selection on Exponential Decay – clarified.
5. Details on Abstention Classification – clarified.
6. Details on the usage of the exact match reward – clarified.

No outstanding unaddressed concerns.


**Reviewer b1mW**:
This reviewer raised seven concerns and they are partially addressed as follows:
1. Theoretical justification on the reward function design – justified in Appendix A and reasonable under the specified assumptions.
2. On the optimal policy behavior in the oracle-helper setting – unsatisfactory as the assumption on the helper is not specified.
3. Generalizability of the key observation on QA with a specific retrieval setup – claimed that it is beyond the scope of our work.
4. Theoretical rationale on the “correctness × search-penalty” proxy for encouraging meaningful “abstention” – no direct answer
5. Empirical details on model’s abstention behaviors – reported but the total abstention rate (not Abs(0) which are already reported) is missing.
6. Effect on direct abstention training explicitly with a ternary reward (e.g., Correct = +1, Abstain = 0, Wrong = −1) – empirically justified that it does not provide expected results (i.e., the training policy is always abstaining).
7. Expand the experiments to more QA datasets and evaluate for generalizability on the observation – claiming that it is beyond the scope of our work.

The generalizability of the proposed method still remains. In particular, the authors' theoretical justification in Appendix A does not provide a clear definition on an optimal policy and depends on Assumption 2, which is not trivial as it depends on a helper. Currently, the proposed abstention method heavily relies on the performance of the helper, so at least authors need to show empirical justification whether the claim maintains over helpers with diverse performance levels, which is missing.

**Reviewer Scores:**

**Reviewer xSsG**:
Final expected rating: 6 / final expected confidence: 4 – This reviewer increased its score to 6 and decreased its confidence to 4 before the discussion shutdown and I think it is reasonable as the most concerns are addressed.

**Reviewer wZVe**:
Final expected rating: 6 / final expected confidence: 2 – This reviewer shared final scores, which is reasonable to me as well.

**Reviewer 2Hdt**
Final expected rating: 6 / final expected confidence: 3 – I expect that the reviewer’s concerns are all reasonably addressed and the reviewer would maintain its score if the discussion had proceeded. But, I cannot find sufficient reasons to increase ratings compared to other papers with 8 scores.

**Reviewer b1mW**
Final expected rating: 4 / final expected confidence: 4 – Critical concerns on the generalizability of the method still remain and it looks harder to address during the rebuttal period. So, I expect that the reviewer would maintain the scores.

---

### Decision · Program_Chairs · 2026-01-26

Reject